# Extreme marine events revealed by lagoonal sedimentary records in Ghar el Melh during the last 2500 years in the northeast of Tunisia.

Balkis Samah Kohila[1, 2], Laurent Dezileau[2], Soumaya Boussetta[1], Tarek Melki[1], Nejib Kallel. [1]

[1] Laboratoire Géoressources, Matériaux, Environnements et changements globaux, LR13ES23 (GEOGLOB), Faculté des Sciences de Sfax, BP1171, Sfax 3000, Université de Sfax, Tunisie.
[2] Laboratoire de Morphodynamique Continentale et Côtière (M2C), Université de Caen, UMR 6143, 14000 Caen, France.

*Correspondence to*: Balkis Samah Kohila (balkis.samah@yahoo.fr)

**Abstract.** The Tunisian coast has been affected in the past by many events of extreme marine submersion (storms and tsunamis). A high-resolution study along two sediment cores taken from the lagoon of the Ghar el Melh was performed to identify the different paleo-extreme events and to reconstruct the paleo-environmental changes of the North-eastern part of Tunisia during the Late Holocene. A very high-resolution sedimentological analysis (granulometric, and geochemical) was applied to these cores. These cores were also dated with isotopic techniques ($^{137}$Cs, $^{210}$Pb$_{ex}$, $^{14}$C) and the outcomes reveal five phases of paleoenvironmental changes of this lagoonal complex and identify two sediment layers that are in connection with two major historical marine submersion events. The first layer is mentioned as E1 and seems to fit with the great tsunami of 365 Cal AD. This event was marked by an increase in the coarse sediment and it is correlated for the first time with the immersed city of Neapolis in northern Golf of Hammamet in 2017 by the same tsunamis of 365 Cal AD. The other sandy layer referred to as E2 was dated from 1690 to 1760 Cal AD, and is marked by one specific sedimentological layer attributed to a marine submersion event. This layer could be associated with the 1693 tsunami event in southern Italy or an increase in extreme storm events.

# 1 Introduction

During the last century, coastal communities have become very vulnerable to many extreme events such as tsunamis, tropical storms, hurricanes, and floods (Chaumillon et al., 2017). Risks and vulnerabilities of the coastal area have recently increased, not only because of the sea-level rise the changes in climate conditions but also because of the high number of natural catastrophes disasters, and the construction of nonplanned urban areas (Cardona A., 2001; Milanés Batista et al., 2017).

However, coastal storms or tsunamis hazards are among the most dangerous events that caused significant human and economic losses along coastal regions (Seisdedos et al., 2013). One of the destructive recorded meteorological events was the hurricane Katrina in Florida in the United States in 2005. It engenders more than $100 billion of damages, killed around 2000 people, and touched 90000 square miles of the United States (Phadke, 2005). The 2004 Sumatra tsunami was consecutive to a mega-earthquake with $M_w$ 9.2. This tsunami has generated high waves up to 30 meters and induced 250.000 dead peoples (Paris et al., 2010). This tsunami was considered the most hazardous event compared to other past catastrophic events that occurred in 1797 and 1833 on the coast of Sumatra island (Ahmadun et al., 2020).

The Mediterranean basin is defined as a "Hotspot" of the climate change (Lionello and Scarascia, 2018; Pausas and Millán, 2019). According to the coastal erosion and submersion are some of the results of an elevation of sea level due to global warming. The Mediterranean Sea surface temperature has increased by almost 1°C since 1980 and is expected to rise further by 2.5 °C in the next seventy years (Jacqueline Karas, 2006). In this context, the Mediterranean coastal zones will be probably more vulnerable to climatic extreme events (storms and medicanes), and more exposed to coastal erosion processes and flooding (Seisdedos et al., 2013). The Mediterranean basin is also characterized by a high seismic activity due to its geographical position between the Eurasian and African plates (Papadopoulos and Fokaefs, 2005; Papadopoulos and Baskoutas, 2009). The earthquake of the 21st of May 2003 that happened in the Western Mediterranean Sub-basin precisely in Boumerdes (Algeria) with a magnitude of about $M_w$ 6.9 has generated a tsunami (Sahal et al., 2009). The wave generated has a height < 25cm and propagated from the Algerian coast toward the Murcia Province. These waves have not caused any evident damages (Álvarez-Gómez and Gonzalez, 2011). However, in the Eastern Mediterranean sub-basin, the Crete earthquake (21st July 365 AD) induced a tsunami that propagated across the region to reach Alexandria and hit the Cretan coast causing extensive damage (Scardino et al., 2020).

The Tunisian coast has been exposed to numerous extreme hazards (floods, storms, and tsunamis) (Rizzi et al., 2016; Zaïbi et al., 2016; Affouri et al., 2017; Khadraoui et al., 2018; Amrouni et al., 2019). During the last century, this coastal area has experienced some coastal marine storms (Zaïbi et al., 2016). Moreover, this area is also subject to tsunami events, which can especially come from the seismic source related to the tectonic activities in Southeastern Sicily e.g the immersed city of Neapolis in northern Golf of Hammamet in 2017 suggest the occurrence of a tsunami in 365 AD (Aounallah and Fantar, 2006; National Heritage Institute of Tunisia, 2017).

Due to the absence of high-resolution data covering a sufficiently long time period, the recent instrumental data, and textual

archive on extreme events in Tunisia do not allow us to determinate any evolution in time. Since these extreme hazards (Storm and Tsunami) are causing many sedimentary inputs in coastal areas, the recent geological records existing in this area will allow the study of these extreme events over a longer time, beyond the textural and instrumental textural archives (Morton et al., 2007; Dezileau et al., 2011; Sabatier et al., 2012). This geological approach using sedimentological and geochemical analyses has been used in the French, Morocco, and Spanish coasts (Degeai et al., 2015; Dezileau et al., 2016; Khalfaoui et

al., 2019) . Inversely, only a few high-resolutions studies have been conducted on the Tunisian coast. In this context, the present study aims to reconstruct past marine submersion events from geological archives (GEM3 and GEM4 cores) collected from the Ghar el Melh lagoon (NE of Tunisia) using a high-resolution sedimentological and geochemical analysis.

## 2 The study site of Ghar El Melh

### 2.1 Geological and geomorphological setting

This work focuses on the Ghar el Melh lagoon situated in the Northeast of Tunisia. This lagoon is also called "the Porto Farina". It has an elliptic shape and approximately a surface of 28.5 km$^2$. Its average depth is about ~0.8 m (Romdhane, 1985; SCET-ERI, 2000; Moussa et al., 2005). This lagoon is directly limited in the north side by a mountain range called "Jbel Nadhour" (325 m). This mountain is composed of a marine Pliocene material (Figure 1) represented by sandstone sediments (Hamouda, 2014). The lagoon is bordered in the west and south side by recent quaternary marshy grounds formed by clay and

silt sediments. In the eastern side, it is separated from the sea by a sandy barrier, with a local opening (El Boughaz) allowing a permanent hydraulic communication (Oueslati et al., 2006). This sandy barrier was formed by a littoral drift oriented from the North-East to the South-West (KHRYSTAL Engineering, 2003).

According to Paskoff (1994), the lagoon was considered as a vestigial and remaining part of the Utique Sea that was formed during the last postglacial transgression since 6000 years ago. Progressively, this small gulf has been disconnected from the

Utique Sea, due to Medjerda fluvial deposits (Paskoff and Trousset, 1992). This caused  a progressive reshaping of the lagoon to its present morphology, which could be attributed to an association of the shape of the coastline and alluvium deposition from the Medjerda River (Moussa et al., 2005). Over time, the delta of Medjerda was distinguished by a high interannual discharge variability. This river's average sediment yield is about 10 g/l and it is characterized by an annual average flow of 30 m3/s and reached 3500 m3/s in the exceptional flood of March 1973 where solid discharge is calculated around 100 g/l

(Claude et al., 1977). A large amounts of alluvium was carried by the Oued, estimated over 22 millions tonnes per year in the Gulf of Tunis before the construction of dams (Oueslati et al., 2006). The sediments are deposited when the Oued reaches the low flood zones, contributing then to an extension of his delta towards the sea and numerous changes of channels (Delile et al., 2012).

In the Gulf of Tunis, the mean amplitude of semi-diurnal micro-tidal activities measures between 12 and 30 cm (El Arrim,

1996; Saïdi et al., 2012) The amplitude of the tidal range in this region was estimated around 35 cm (Oueslati, 1993). The

coastal environment of the Gulf of Tunis was exposed to natural erosion processes induced by waves, tides, and periodic storm-surges (Hzami et al., 2021). This erosion is the effect of the long shore coastal drift from the south-east to north-west direction.

## 2.2  Climatological and hydrological settings

The Northeast of Tunisia is characterized by a typical Mediterranean climate with arid summer and rainy winter characterized
generally by heavy rainfall periods and floods. The rainfall mean annual in the lower valley of the Medjerda is around 500 mm.yr$^{-1}$ (Oueslati, 2004). Furthermore, in this study area, the rainfall rate is very variable. The mean winter temperatures are around ~11 °C. The highest precipitation occurs mainly between November and December with an average of 248 mm (INS, 2001). The monthly average values sometimes exceed 100 mm (Beni Atta Station in December) and never fall below 60 mm (MEAT, 2001). The high temperatures are observed in August with mean values of ~27 °C. In other hand, the  salinity in the
lagoon depends on the hydrological balance and varies from 36 g/l in winter to 51 g/l in summer due to the higher evaporation rate (1450 mm/year) (Added, 2001; Moussa et al., 2005; Oueslati et al., 2006).

## 3 Material and methods

### 3.1 Sampling location and sediment samples

Two piston cores were manually collected in 2012 in the Northeast of Ghar el Melh lagoon. These cores are 126 cm (GEM4)
and 98 cm (GEM3) in length and 10 cm in diameter (Figure 1). They were manually sampled according an East West transect into the lagoon (~200m from the sandy barrier for GEM3 and ~400m for GEM4). In the laboratory, the two cores GEM4 and GEM3 were photographed and described in detail. Before granulometric and geochemical analysis, the cores were split into 1 cm vertical sections. Moreover, 29 surface sediment samples of around 20 to 30g were collected from present-day soil horizons from the Medejerda watershed to the littoral area (beaches and dunes) to assess the origin and all sources of the lagoon sediment
arriving in the study area: (i) 6 samples comes from the Medjerda River (Gr01 to Gr06), (ii) 12 samples from different small affluents located in the northern and western part of the lagoon (from Gr07 to Gr18) and (iii) 11 samples from the sandy barrier (Gr19 to Gr29) (Figure 1). In the laboratory, granulometric and geochemical analysis were performed on these surface samples. The particle sizes obtained are classified according to Folk and Ward (1957) into three categories (clay $\Phi$ <2µm, silt 2µm <$\Phi$ < 63µm and sand $\Phi$ > 63µm).

### 3.2 Sedimentology and geochemistry analysis

To determine the distribution of grain size a particle size analysis was adopted by using a Beckman Coulter© LS 13 320 (Geosciences Montpellier). Due to the high concentration of shells fragments (>200 µm), every sample was sieved at 150µm before analysis. An ultrasounds were used to avoid particles flocculation, after the entrance of sediments into the fluid module of the device. The elemental geochemical analyses by energy-dispersive X-ray fluorescence (XRF) spectrometry were
undertaken on sediment cores and surface samples with a hand-held Niton XL3t. In order to avoid desiccation of the sediment

and contamination of the XRF measurement unit, samples had to be covered with an Ultralean film. The geochemical analysis from XRF measurements was executed in the mining type ModCF proline mode. These semi-quantitative elemental measurements were performed along with the sediment core every 1 cm. The elemental concentrations obtained in this work using the hand-held Nitron XL3t are expressed in ppm or percentage values.

**3.3 Chronological framework**

The chronology of the two cores for the last century was measured using $^{210}$Pb$_{ex}$ and $^{137}$Cs measurements with a CANBERRA Broad Energy Ge (BEGe) detector on fine sediment (fraction < 150 µm). The $^{210}$Pb$_{ex}$ dating is founded on the determination of the $^{210}$Pb excess activities preserved in the sediment of cores. The use of this natural radionuclide $^{210}$Pb to indicate sedimentation rate is now a well-established technique (Goldberg, 1963; Krishnaswamy et al., 1971; Robbins and Edgington, 1975). The dating of $^{137}$Cs was released following the procedure of Robbins and Edgington (1975). To complete the chronology of the two cores over longer periods of time, $^{14}$C analyses were performed on mollusk species (*Cerastoderma glaucum*) at the laboratoire de Mesure C14 (LMC14) on the ARTEMIS accelerator in the French CEA (Atomic Energy Commission) at Saclay, France. In Fact, these $^{14}$C analyses were realized with the classical procedure illustrated by Tisnérat-Laborde et al., 2001. The radiocarbon ages were transformed into calendar ages utilizing the Marine 13 curve (Reimer et al., 2013). The radiocarbon ages of marine and lagoonal organisms are generally older than the atmospheric $^{14}$C ages and have been calculated and modified by subtracting the « reservoir age » (Zoppi et al., 2001; Siani et al., 2001; Reimer and McCormac, 2002; Sabatier et al., 2010; Dezileau et al., 2016).

## 4  Results

### 4.1  Downcore results

#### 4.1.1 Lithological, granulometric and geochemical studies of GEM3 and GEM4 cores

Two sedimentary cores were collected in the northeastern part of Ghar el Melh lagoon. They contain fine sediments (clay and silt) interbedded with coarse-grained layers formed by mollusk fragments and siliciclastic sand. The sedimentary succession in the GEM3 core is very comparable to that presented in GEM4 (Figure 2). The thickness of sandy layers is slightly different between the two cores. This could be attributed to the geographical position of core GEM3, which is closer to the sandy barrier (~200m) than core GEM4 (~400m).

The GEM3 and GEM4 sediment cores of 97 and 126 cm long show visual variation in the sediment composition. Lithological description of these two cores highlighted five distinct sedimentary facies (Figure 2):

The first unit (1), situated between 126-67 cm in GEM4 and between 97-85 cm in GEM3, is composed generally of a light grey silt layer and shells. At the base (the last 3 cm for the GEM3 and the last 13 cm for the GEM4) the lithological composition of this unit is characterized by a very thin fine sand. For the GEM3, the transition between unit (1) to unit (2) is defined by a

sharp contact (Figure 2).

The second unit (2), situated between 85-63 cm in GEM3 and between 66-60 cm in GEM4, is typically composed of a light grey sand with a combination of shell fragments and siliciclastic grains. It is probably related to marine incursion and washover event during an intense event such as a storm or tsunami.

The third unit (3), situated between 60-36 cm in the GEM3 and between 63-30 cm in the GEM4, is composed mainly of grey silt and shells. In fact, the transition between this unit and the unit subjacent is defined by a discontinuity contact (Figure 2).

The fourth unit (4), is about 10 cm think in GEM3 and 6 cm in the GEM4. This coarse layer is constituted by a mixture of shell debris and siliciclastic sand. This sand layer is usually characterized by coarse sediments with light colours and also dominated by shell fragments. This coarse grain size layer intercalated in the mud sediments indicates an "energetic" event, relative to the background sedimentation. It is probably also linked to washover event and marine incursion during an intense
event such as a storm or tsunami.

The fifth unit (5) presenting a thickness of 26 cm (GEM3) and 24 cm (GEM4) and marked by a massive grey to dark clay with trace of oxidized plant roots in the last three centimeters.

The granulometric analysis shows that the silty clay is the most abundant fraction in our sediment cores (Figure 2). However,
sand is dominant at two levels. The first layer is observed between 86 and 64 cm in GEM3 and between 60 and 65 cm in GEM4 whereas the second one is between 35 to 26 cm in GEM3 and from 30 to 24 cm in GEM4. These two layers are characterized by the dominance of a sandy quartz material and shell fragments.

The geochemical analysis of GEM3 and GEM4 cores performed using an XRF core scanner, have detected 21 chemical elements above the limit of detection. Among these elements, we choose to represent in Figure 2 only those with a significant
down core variation which are Si, Ca, Sr, and Ti. The Silicon will represent the marine geochemical pole, while Titanium represents the terrigenous one. According to the granulometric and geochemical results of surface samples, the Ti is associated with the silt-clay fraction whereas the "Si" is associated with the sandy fraction. Furthermore, the highest Strontium Sr and Calcium Ca values in the two cores are found to be related with the silt-clay fraction between 62 and 33 cm in GEM3 and between 60 and 30 cm in GEM4.

**4.1.2 Age model**

The chronology of the GEM3 and GEM4 cores has been established using the $^{14}C$ dates and also the $^{137}Cs$ and $^{210}Pb$ measurements on monospecific mollusk shell samples and bulk sediments, respectively. In the uppermost 30 cm of the two cores, the measured $^{210}Pb_{ex}$ values range in GEM3 from 287.47 to 2.29 mBqg$^{-1}$ and f in GEM4 from 253.67 to 1.01 mBqg$^{-1}$. In general, the down-core distribution of $^{210}Pb$ excess values follows a relatively exponential decrease with depth. Based on a
model Constant Flux, Constant Sedimentation Rate (CFCS) (applicated by Goldberg, 1963; Krishnaswamy et al., 1971), the $^{210}Pb$ data determinate a sedimentation rate of 0.7 mm year$^{-1}$ for GEM3 and 0.6 mm year$^{-1}$ for GEM4. The distribution profile

of [137]Cs activity shows for the first 10 cm (Figure 3) a maximum value at 3.5 cm in GEM3 and 3 cm in GEM4. This can represent the period of the utmost radionuclide fallout in the Northern Hemisphere which was related to the weapons atomic peak testing in 1963. The [137]Cs obtained sedimentation rate is about ~0.7 mm year$^{-1}$ for the GEM3 which is slightly higher

than that obtained for GEM4 (0.6 mm year$^{-1}$; Figure 4). The distribution profile of the total Pb shows that the beginning of the industrial pollution in 1892 (Latour, 2019) is situated at 9 cm, indicating a sedimentation rate of about 0.75 mm year$^{-1}$ for both cores (Figure 4). Thereby, sedimentation rate of the GEM3 core calculated using the [137]Cs, [210]Pb and total Pb show similar values around 0.75 mm year$^{-1}$, which are slightly higher than that for GEM4 (mean value of around of ~0.6 mm year$^{-1}$). The conventional AMS-[14]C measurements were performed using mollusk shells (*Cerastoderma glaucum*) on eight control points

for GEM3 (Table 1) and seven ones for the GEM4 (Table 2). Taking into account both the radiocarbon and [210]Pb$_{ex}$ dates, the local [14]C reservoir age in the Ghar el Melh lagoon was determined. According to the method of Sabatier et al.(2010), the evaluation of the modern [14]C reservoir age was conducted by comparing an age obtained from [137]Cs and [210]Pb data, and from geochemical analysis of mining-contaminated lagoonal sediments with an AMS[14]C age of a pre-bomb mollusk shell.

According to Reimer et al., 2013, the Sea surface reservoir age R(t) for the recent shell (SacA44506) was measured by

subtracting the atmospheric [14]C value determined for the historical date 1845 AD (114±8 [14]C years) from the measured apparent [14]C ages of the shell (450±30 [14]C years, Table 3). This determines an R(t) value of 363 years. The deviance from the total mean reservoir age (ΔR) is then calculated by subtracting the marine age model value obtained for 1845 AD (488±23 [14]C years) from the measured apparent [14]C age of the mollusk (450±30 [14]C years; Table 3). The calculated ΔR around -38 years (Table 3) is thus, adopted. Finally, the age model of GEM3 and GEM4 cores was established by using the OxCal 4 on [14]C ages

and [210]Pb$_{ex}$/[137]Cs average sedimentation rates. The [14]C mean sedimentation rate calculated is then about ~2.5mm year$^{-1}$ (Figure 5A).

As the GEM4 indicates a level of erosion or inactive deposition process named "a Condensed area" observed from 59 to 72 cm (between 0 and 1000 Cal AD/BC), which would instigate an error in the obtained age model (Figure 5B), the discussion will be focused only on the ages estimated from GEM3 core. For the GEM4 core, the age correction has been obtained by the

correlation between the Strontium profile of the two cores GEM3 and GEM4. The age models takes into account the depth and the thickness of every event deposits.

## 4.2 Characterization of different detrital surface sources

The sedimentation in the lagoon of the Ghar el Melh is manifested by marine and terrestrial inputs. The geographic distribution (Figure 6) of granulometric results indicates that the high percentage of coarse sediments "sands" (% >75%) are from the sandy

barrier, whereas, sediments of the Medjerda watershed are distinguished by a very high percentage of fine sediments (Silt and Clay). Medjerda River and the affluent around it constitute the main origins of fine fractions (Clay and Silt) in the Ghar el Melh lagoon. The mapping of terrigenous elements such as Silicon, Titanium, and Iron contents in surface sediments confirms this distinction of detrital origins around the Ghar el Melh lagoon. High Si values (>110000 ppm) distinguish especially the

sandy barrier. Moreover, the highest contents of Titanium Ti (> 1400 ppm) and Iron Fe (> 17500 ppm) are retrieved in sediment from the watershed of the Medjerda (Figure 7A). To make our interpretations more vigorous, a tree diagram was generated using the statistical program XLSTAT 2021 statistical software, which is used as an additional tool to identify and test the statistical link between all the elements and the deposits using both sedimentological data and XRF data of surface sediments in the study area (Figure 7B). In the first cluster, the association of coarse fraction (Sand) with the Si is clear, suggesting that the silicone is coming from coarse marine sand inputs. However, the second cluster determines an assembly between the terrigenous elements (Ti, Fe, Sr, and Ca) and fine fractions (Silt and Sand). This difference in the origin of the terrigenous inputs in Ghar el Melh lagoon explained by the fact that, during floods events, finer sediments are coming from the Medejerda watershed whereas, at the time of marine storms, coarse marine sand inputs are from the barrier.

In coastal environments, the principal component analysis (PCA) was usually performed on the sediment sampled around lagoons in order to characterize the different sources of sediments deposited in the lagoon (Degeai et al., 2015; Gaceur et al., 2017; Affouri et al., 2017 and Khalfaoui et al., 2019) and to determine the several end-membres related to potential sediment sources supplies (Figure 8). We choose the Mn, Ti, Zn, Ba, Rb, Fe, Sr, Ca, and Si elements due to their good detection by the mobile XRF. We established the calculation factors F1 61.12% and F2 11.13% of the geochemical dataset using the XLSTAT-2016 statistical software. The two first factorial factors represented in the PCA diagram present (Figure 8) 72.25% of the entire variance of the dataset. The factor 1 accounts for 61.12% of the entire variance. Factor number 1 is marked by a positive loading for terrigenous elements Rb, Ti, Ba, Mn, Fe, and Zn. Whereas the Ca and Sr present a modest positive loading and are inserted in factor 1. Factor 2 represents 11.13% of the global variance. It indicates positive loading for Si, Mn, Ti, Ba, Zn, Rb, and Fe, whereas Sr and Ca have negative loadings.

The geochemical results of the downcore sediments (CEM3 and CEM4) show the variation in the concentration of chemical elements such as Fe, Ti, Ca, Sr, and Si. The PCA (Principal Component Analysis) of the two cores shows the presence of three well-differentiated poles (Figure 8). The F1 and F2 statistical variables reveal a good correlation between the elements present in the same pole (89% for the GEM3 and 79% for the GEM4).

Based on these statistical analyses of all geochemical data (surface sediments and cores), three distinct sources of sediments were identified (Figure 8): (i) Terrigenous or Alluvial source (Mn, Fe, Zn, Ba, Rb) mainly discharged by rivers during floods. (ii) Marine source Si such as sands coming from the sandy barrier coastal during marine submersion; (iii) and the Sr and Ca represent an authogenic and/or biogenic origin linked to the precipitation of minerals or the dissolution of some shells.

## 5    Discussion

### 5.1 Site sensitivity to overwash deposits

The sensitivity of the site to the overwash deposits can result from several factors such as barrier elevation, sediment supply, inlet, and a change in sea level (Donnelly et al., 2004; Scileppi and Donnelly, 2007; Dezileau et al., 2016). Generally, an increase in sea level produces a moving of the barrier landward. Thus, the highest number of sand layers in a sediment core

can be the consequence of a simple sea-level change. In the Mediterranean Sea and especially during the last 5000 years, the sea level has stayed more or less stable (< 2 m, Pirazzoli, 1991; Lambeck and Bard, 2000).

Studies concerning Holocene sea-level fluctuations along the Tunisian coast have suggested a stabilization in relative sea level during the last 6000 years (Jedoui et al., 1998), which is very small and probably not enough to completely change the deposition environment of the Ghar el Melh lagoon. Furthermore, sedimentation in the two cores started 2500 years ago, on this short period, the influence of the seal level change has not drastically affected sedimentation in the center of the lagoon.

Cores GEM3 and GEM4 present generally a sedimentary sequence dominated by fine-grained sediments (clay and silt), suggesting that the lagoon of Ghar el Melh has succeeded in keeping a low-energy environment during the last 2500 years.

The proxies applied in this work (granulometry and geochemistry), as well as the sedimentary indicators (discontinuity contact), showed that all sandy coarse peaks present within the GEM3 and the GEM4 cores were deposited through marine high-energy events and not by a gradual change in sea-level.

The presence of a nearby inlet may increase the sensitivity of a particular area to storm-induced deposition. It allows for a lesser storm with a lower wave surge to more easily penetrate and transport coarse sediment into the back-barrier area. If a large inlet had existed over a long period and had provided a ready conduit for sand from the Gulf of Tunis to the Ghar el Melh lagoon, this would have been reflected in the cores (deposition of thick sand layer over a long period of time). However, no evidence of such active tidal connection lasting a long time is found in sedimentological and geochemical data for the past 2500 years.

Granulometric and Geochemical observations on the GEM3 and the GEM4 cores show a sedimentary sequence dominated by silty-clay deposits and interrupted over time by some allochthonous coarse materials. To determinate the source of these sandy layers, we compared their geochemical attributes with those from surface samples. This allowed us to establish whether these deposits had a continental (river floods) or marine origin (storms and tsunamis). The two sand deposits present in the GEM3 and GEM4 cores have a geochemical correlation with marine coastal surface samples; both show enrichment with Si and depletion with Fe and Ti, which reveals a marine source for these high-energy deposits. The results demonstrate that Ghar el Melh lagoon has been confronted with different episodes of marine submersion during the last 2500 years. In this respect, the lagoonal deposit of Ghar el Melh can provide valuable information on these aspects of the past and subsequently provide a forecast about the future. So we can suggest that the Tunisian coast is very sensitive to extreme events, especially these coastal areas are vital for Tunisia's tourism development and economy. These hazard events can expose in the future many destructions and caused significant human and economic losses. In this fact, many managements to risk should be taken into consideration and applied by the governorate. In this fact, many managements to risk in this coastal zone should be taken into consideration and applied by the governorate. A Regional Risk Assessment methodology must be developed for the assessment of the potential impacts of climate change in the Tunisian coastal zone of the Ghar El Melh lagoon.

## 5.2 Extreme events and paleo-evironement changes

The Late Holocene lagoonal history of Ghar el Melh may be divided into five phases that record the connection between the lagoon and the sea in relation to the sandy barrier's evolution. The different phases can be described as follow:

The first phase, dated from -275 to 300 Cal AD, is marked by a high percentage of fine sediments (about 70 % of Silt) (Figure 9). Our geochemical results show a relatively high concentration of terrigenous elements Ti (around 250 ppm) and Fe (around 2500 ppm). This predominance of the fine fraction, rich in Ti and Fe, means that the lagoon is filled in sediments coming from the Medjerda River. The presence of fine material suggests that the hydrodynamic current in the lagoon is low, we have a protected lagoon with a sandy barrier well-constructed. This accumulation of fine sediment could also indicates a higher contribution of sediments from Medjerda River. Indeed, this interval (Roman Climatic Optimum) corresponds to a more humid phase in the North of Tunisia (Stevenson et al., 1993).

The second phase started around 300 Cal AD and finished at 1100 Cal AD. Our granulometric and geochemical results demonstrate that the sedimentation of Ghar el Melh lagoon during this period was mainly controlled by a marine contribution. This phase is marked by a decrease in the percentage of silt, and a rise in the percentage of sand (about 80 %) and the presence of a discontinuity contact with the underlying unit. This time interval shows also the presence of a high amount of Silicon (Si >100000 ppm) (Figure 9) which stipulate an increase in the sandy material supply. The decrease in the silty-clay fraction (10%) can be related to a dilution by high marine inputs. The dominance of coarse sediment, rich in Si, could be explained by a weakening of the sandy barrier due to an increase of storm events. Degeai et al.(2015) and Sabatier et al.(2012) have clearly recorded a period of higher storm activity from 400 to 800 cal yr AD in the occidental part of the Mediterranean area. This period named Dark Age Cold Period coincides with the North Atlantic cooling phase known as Bond Event 1 (Bond et al., 2001). Dezileau et al.(2011) and Sabatier et al.(2012) have demonstrated that intervals of an increase in storm activity in the Western Mediterranean area seem to be well related to the cold periods of the Holocene. During these cold periods, the sea ice was prolonged over the North Atlantic basin southward especially during winter (Lamb, 1995). Dezileau et al. (2011, 2016) hypothesized that during the Little Ice Age, the increase of the super-storm activity was probably due to the thermal gradient increase leading to enhanced lower tropospheric baroclinicity over a large Central European domain. This mechanism related to a southward movement of storm track suggests an enhance of storm activity in the Western Mediterranean Sea, in accordance with simulation (Raible et al., 2007). However, this sand deposit may have another origin. Indeed, this deposit was dated at around cal AD 332+/-30. This event E1 coincides with the tsunami event of 365 AD. This extreme event was generated by an earthquake of 8.3 magnitudes (Paris et al., 2020) and is supposed to have been the powerful ever in the Eastern Mediterranean. From a numerical modeling, Pararas-Carayannis and Mader, (2010) indicate that the 365 AD tsunami heavily affected coastal areas throughout the Eastern Mediterranean region; Palestine, South Asia Minor, Cyprus, the Nile Delta, Careen, Apollonia. In the central part of the Mediterranean region the cities of Eastern Sicily, the coastline of Calabria, and the islands of Aiolou were affected (Pararas-Carayannis, 2011). In Tunisia, the recent archeological discovery of the immersed city of Neapolis in northern Golf of Hammamet in 2017 suggest the occurrence of a tsunami in 365 AD (Aounallah and Fantar, 2006; National

Heritage Institute of Tunisia, 2017). We can thus hypothesize that the sand deposited around 365 cal AD could also be associated to this tsunami event. However, the distinction between these two coarse of storm or tsunami deposits is still

controversial and several studies have pointed out many hypotheses regarding the diagnostic characteristics of these deposits (Kortekaas and Dawson, 2007; Morton et al., 2007; Tappin, 2007; Engel et al., 2010; Sakuna-Schwartz et al., 2015). Hence, the sedimentary characteristics of tsunamis or storm deposits are almost similar (Costa et al., 2015). Nevertheless, Morton et al., 2007, used some sedimentological criteria to distinguish storms from tsunami deposits. For example, the storm-originated deposits present a moderately thick sand bed composed of several sub-horizontal planar laminations organized into multiple

laminates. The stratification associated with bed-load transport and abundant shell fragments organized in laminations also favors a storm origin. In contrast, the presence of internal mud laminae or mud intraclasts is stronger evidence of tsunami deposits. However, in our case, the sand bed that corresponds to the extreme event is characterized by a single homogeneous bed (6–9 cm thick) with no evident sedimentary structures (such as laminations) that correspond neither to storm nor tsunami deposits. To precise the origin of our thin coarse layer, we explored the regional historical storm's records and tsunamis data.

The third phase, dated from 1100 to 1690 Cal AD, is marked by a decrease of sandy material and an increase of clay material (about 30%) (Figure 9). The concentration of terrigenous elements, Ti (around 400 ppm) and Fe (around 3500 ppm) (Figure 9) are high during this period. This kind of sedimentation is typically associated with the processes of decantation in the lagoon (Liu and Fearn, 2000; Donnelly et al., 2004). During this third phase, the lagoon is protected by the sandy barrier. This increase in the silty-clay fraction could also be due to a decrease of storm activity and/or an increase of fine sediments transported into

the lagoon by higher runoff of the Medjerda river during the Medieval Warm period.

The fourth phase started at 1690 Cal AD and finished at 1760 Cal AD. This phase was characterized by a high deposition of sand, about 75 % of the total sediment (Figure 9), indicating an opening of the lagoon (Figure 10). The presence of coarse sediments can be explained by a higher marine influence. Interestingly, this coarse-grained layer E2 is recorded in both cores (from 26 to 35 cm for GEM3 and from 24 to 30 cm for GEM4) collected at 200 m and 400 m from the sandy barrier respectively

(Figure 10). Thus, this period could be a sign of an increase of intense storm activity. Indeed, this interval corresponds to the historic period called "Little Ice Age" (LIA) which coincides with the North Atlantic cooling phase known as Bond Event 0 (Bond et al., 2001). Many studies (Sabatier et al., 2012; Degeai et al., 2015; Dezileau et al., 2016) have clearly recorded a higher storm activity between 1400 and 1800 cal yr AD in the occidental region of the Mediterranean Sea. This sand deposit could also be associated to a tsunami event. This period corresponds to the documented Sicilian tsunami that occurred in 1693

CE. Due to its geographic position, the Ghar el Melh lagoon may be affected by this 1693 tsunami.

The sedimentary characteristics of tsunamis or storms deposits are almost similar (Hawkes et al., 2007; Morton et al., 2007; Kortekaas and Dawson, 2007; Mamo et al., 2009). Regarding the diagnostic characteristics of storm or tsunami deposits, several studies have pointed out many hypotheses and determinate that their distinctions are very controversial (Tappin, 2007; Engel and Brückner, 2011; Sakuna-Schwartz et al., 2015). Considering the available data on GEM3 and GEM4 sediment cores,

the hypothesis of tsunami or storm origin for this recorded extreme event remains open. Deciphering the origin of this event

requires further investigations, such a modeling of tsunami wave propagation for example.

The fifth phase, dated from 1760 to 2012 Cal AD, is characterized by fine sediments (Figure 9). This fifth phase shows a protected lagoon (Figure 10) and suggesting that no catastrophic intense sea events have struck Gar El Melh lagoon during this last phase.

**6 Conclusion**

To the paleo-extreme events (storms or tsunamis) and to reconstruct the paleo-evolution of the lagoon of the Ghar el Melh, in the Northeast of Tunisia, high resolution of sedimentological, geochemical and geochronological analysis were used. This approach gives information about the paleoenvironmental changes of Ghar el Melh lagoon and paleo-extreme events since 2300 years. Five phases and two extreme events have been identified: The first phase (from -275 to 300 Cal AD) indicates a
protected lagoon. The second phase from 300 to 1100 Cal AD shows an opening lagoon and more marine inputs. This period was may be associated with a storm activity during the Dark Age Cold Period but may also be related to the 356 AD tsunami event E1. The third phase indicates an isolated lagoon and coincides with the Medieval Warm period (from 1100 to 1690 Cal AD). The fourth phase present an opening phase of the lagoon and may be due to an enhance of storm activity during the Little Ice Age dated or the occurrence of the 1693 tsunami event mentioned in this study as E2. The fifth phase covered the last 250
years and a reclosing lagoon. Even if our records allowed to detect past extreme events, it is not possible to differentiate if they are due to tsunamis or storms. Deciphering the origin of these events requires further investigations.

**Author contributions**

All authors analyzed the results and prepared the manuscript.

**Competing interests**

All authors announce that no competing interests are present.

**Acknowledgements**

The authors would like to thank the Institute at Saclay (French Atomic Energy Commission) and specially the Laboratoire de Mesure $^{14}$C (LMC14) ARTEMIS at the CEA for the $^{14}$C analyses. We are grateful for the projects MISTRALS PALEOMEX,
the PHC-UTIQUE no. 14G1002 and the MEDYNA FP7-IRSES 2014-2017, for your financial support. Also, we would like to thank Radewane Hout for his help in the achievement of some figures.

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

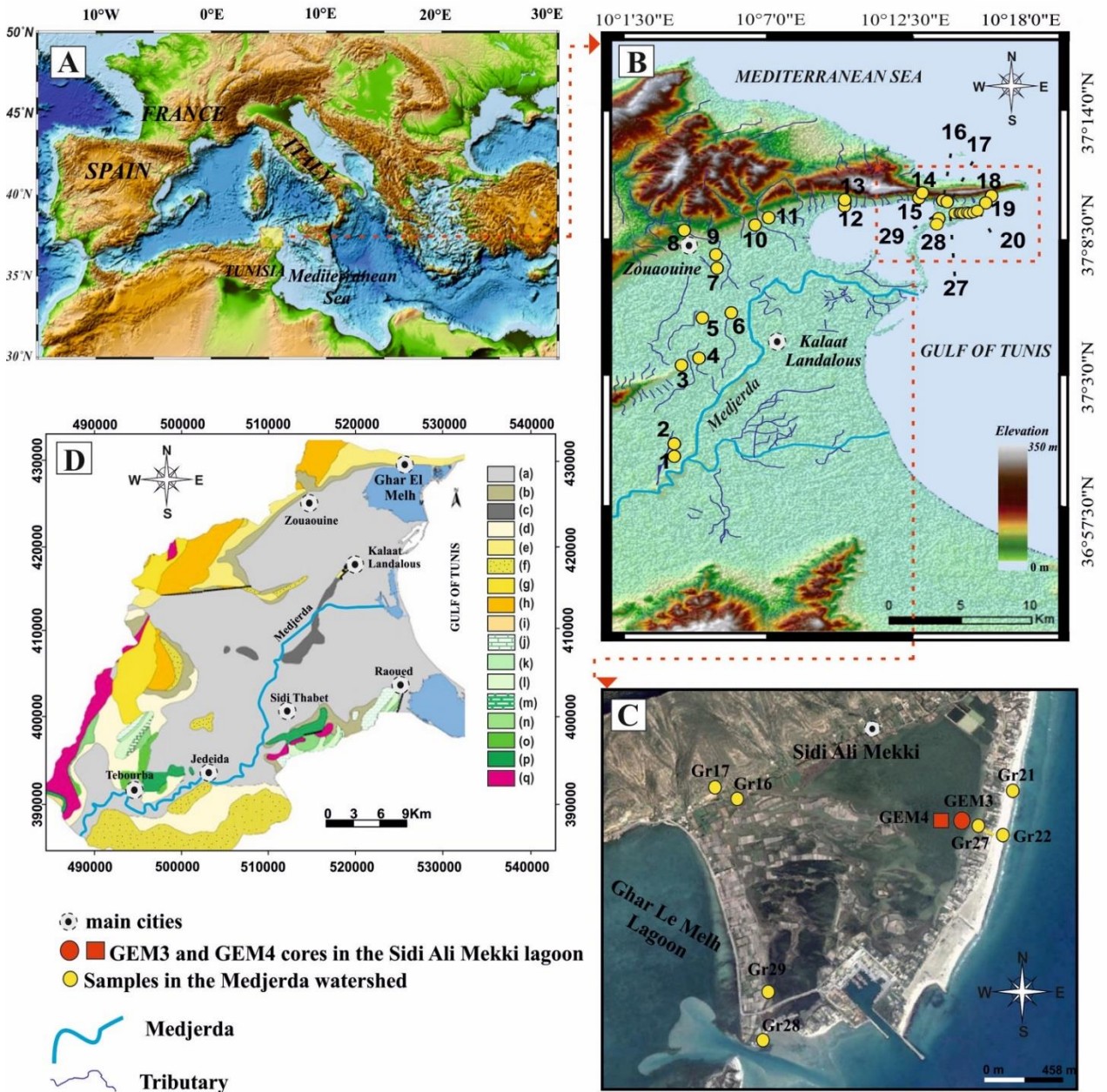

**Figure 1: Maps of geographic setting of the Ghar el Melh Lagoon. (A) Position of the studied region in the Northeastern Mediterranean (Ocean Data View 2013). (B)Topographic map of the lagoon and the emplacement of the different surface sediment samples preleved from the coastal plain of the Medjerda (Created by Radewane Hout using Arcgis). (C) Topographic map of the Ghar el Melh lagoon and its surroundings (e.g ©Google Earth). (D) Geological maps of the Medjerda watershed ((*a*) *Recent alluvial soils (b) Slope deposits (c) Marine Quaternary (d) Villafranchienen (e) Pliocene (f) Continental Pliocene (g) Upper Miocene (h) Flyche de Kchabta (i) Lower Eocene (j) Maastrichtian (k) Senonian (l) Cenomanian (m) Albian (n) Aptian (o) Barrenian (p) Valanginian-Hautrevian (q) Triassi))* (Geological Map; Paper n ° II of Bizerte at 1/200000 modified from Samaali Hamouda, 2011).**


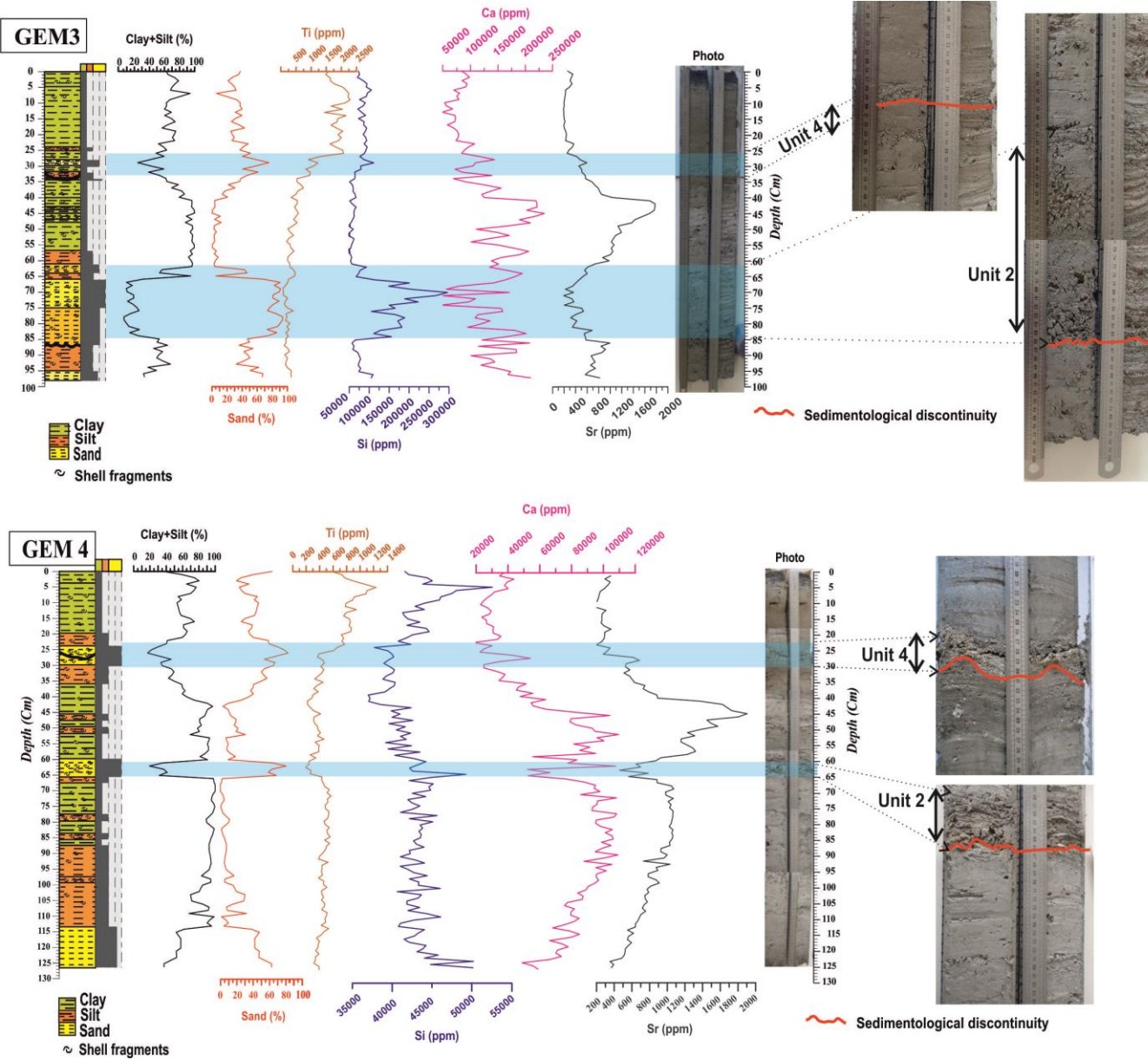

**Figure 2: Stratigraphic log and grain size results of GEM3 and GEM4 cores compared to the results obtained from the XRF records (Titanium, Silicon, Calcium and Strontium) distributing in the two cores, vs. depth.**

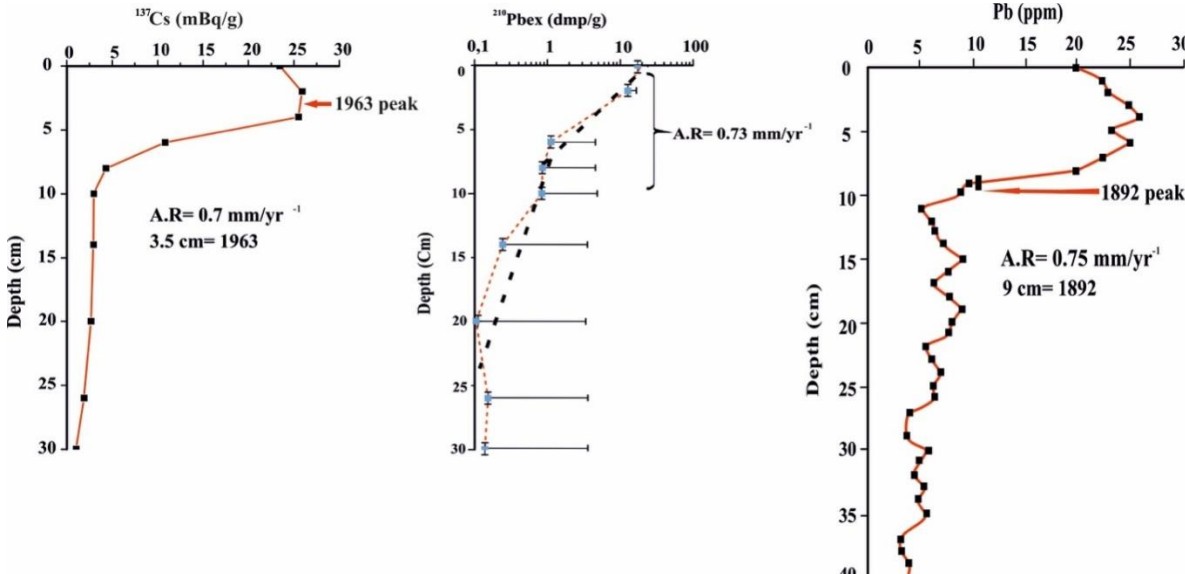

**Figure 3: Distributions profile of $^{210}$Pbex, Pb and $^{137}$Cs vs. depth in the GEM3. The $^{210}$Pb data determinate a sedimentation rate of 0.73 mm year$^{-1}$. The activity depth profile Cs present a peak at 3.5 cm, indicating in accumulation rates of 0.7 mm year$^{-1}$ for the 1963 depths.**


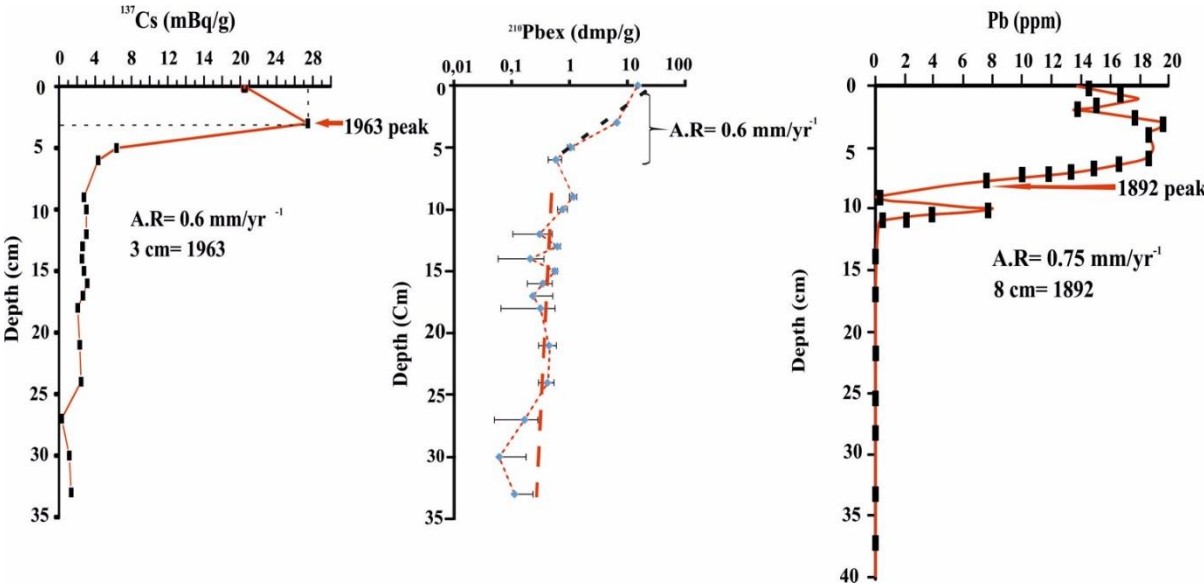

**Figure 4: Distributions profile of $^{210}$Pbex, Pb and $^{137}$Cs vs. depth in the GEM4. The $^{210}$Pb data present a sedimentation rate of 0.6 mm year-1. The activity depth profile Cs indicate a peak at 3 cm, determining in accumulation rates of 0.6 mm year$^{-1}$ for the 1963 depths.**


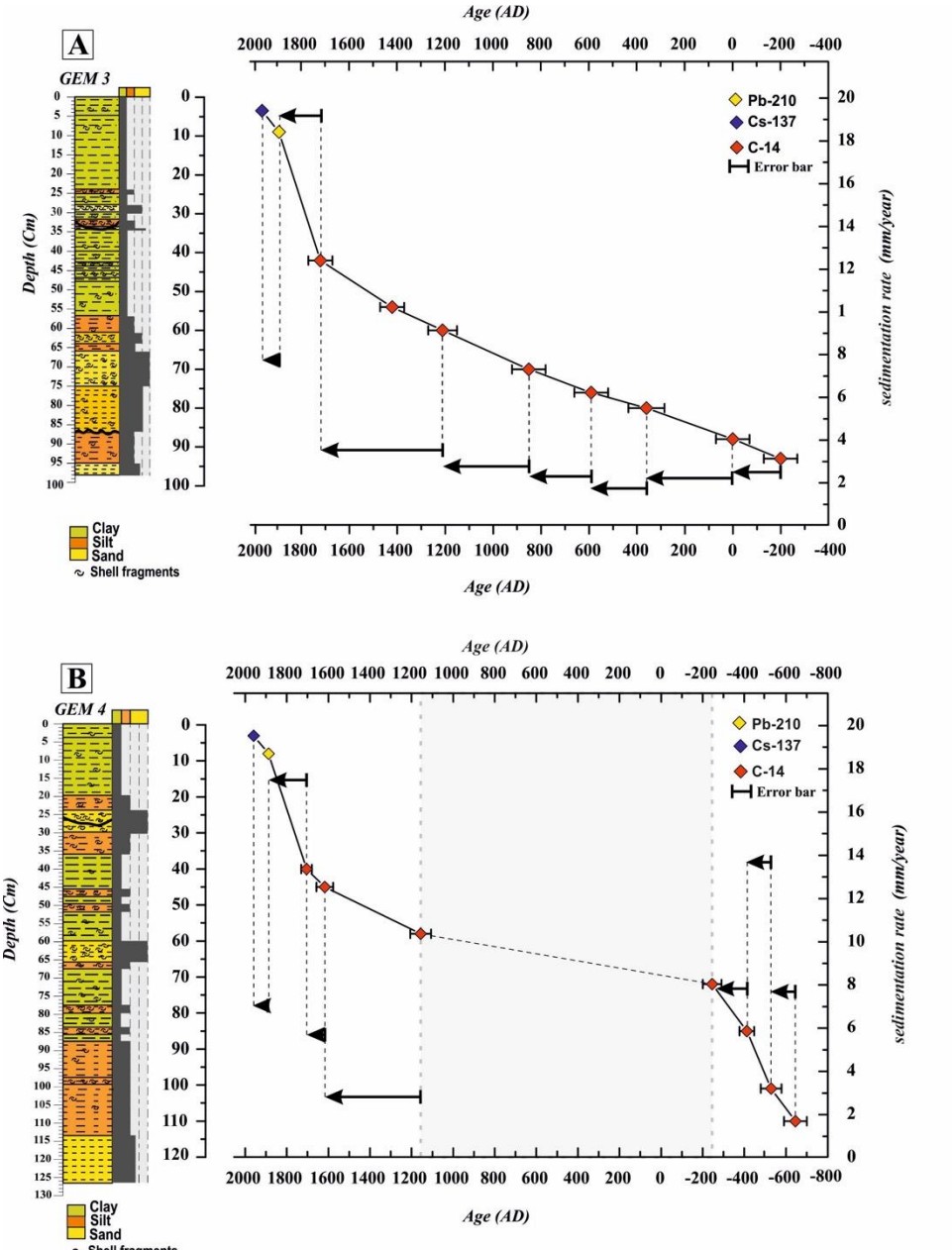

**Figure 5: (5A): Sediment age-depth for the core GEM3 sampled from Ghar el Melh lagoon (Arrows: sedimentation rates variation). The OxCal 4.3 have been used to calculated the age model with 8 samples for [14]C dates and only one sample for each Pb and Cs dates. (5B): Sediment age-depth for the core GEM3 sampled from Ghar el Melh lagoon (Arrows: sedimentation rates variation). The OxCal 4.3 have been used to calculated the age model with 7 samples for [14]C dates and only one sample for each Pb and Cs dates. The blue band determines a period of inactive deposition process or erosion so-called "a Condensed ar**

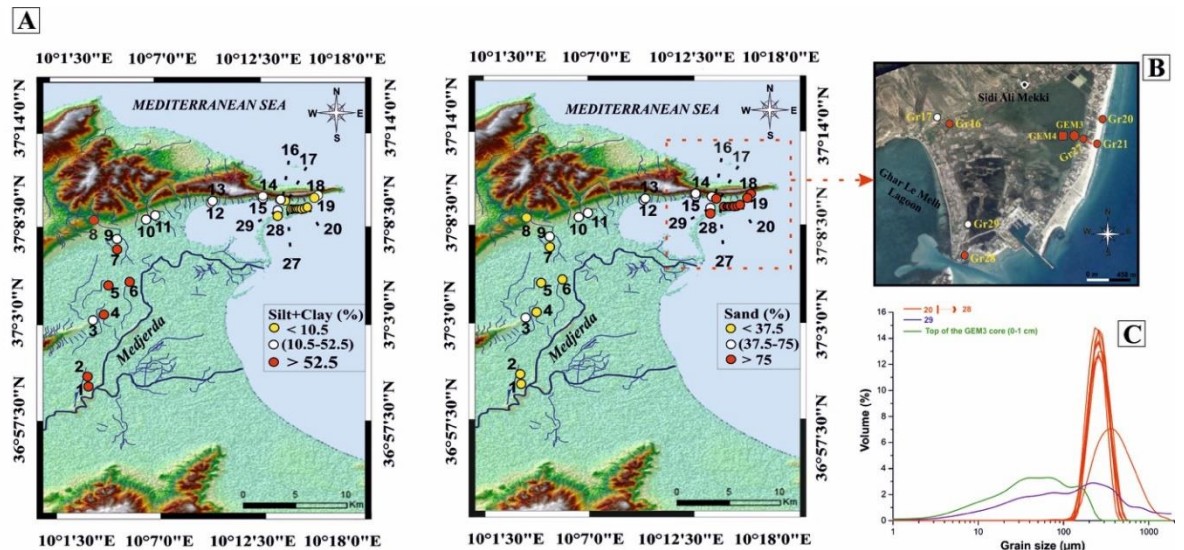

**Figure 6: (6A): Topographic maps of the sand, Silt, and Clay percentage distributed in the Medjerda watershed and around the lagoon of the Ghar el Melh (Created by Radewane Hout using Arcgis). (6B) Topographic map of the two cores and the different sediment samples collected from Ghar el Melh lagoon and its surroundings (e.g © Google Earth). (6C): Particle size distributions (φ < 2000 μm) of representative samples around the Ghar el Melh lagoon.**

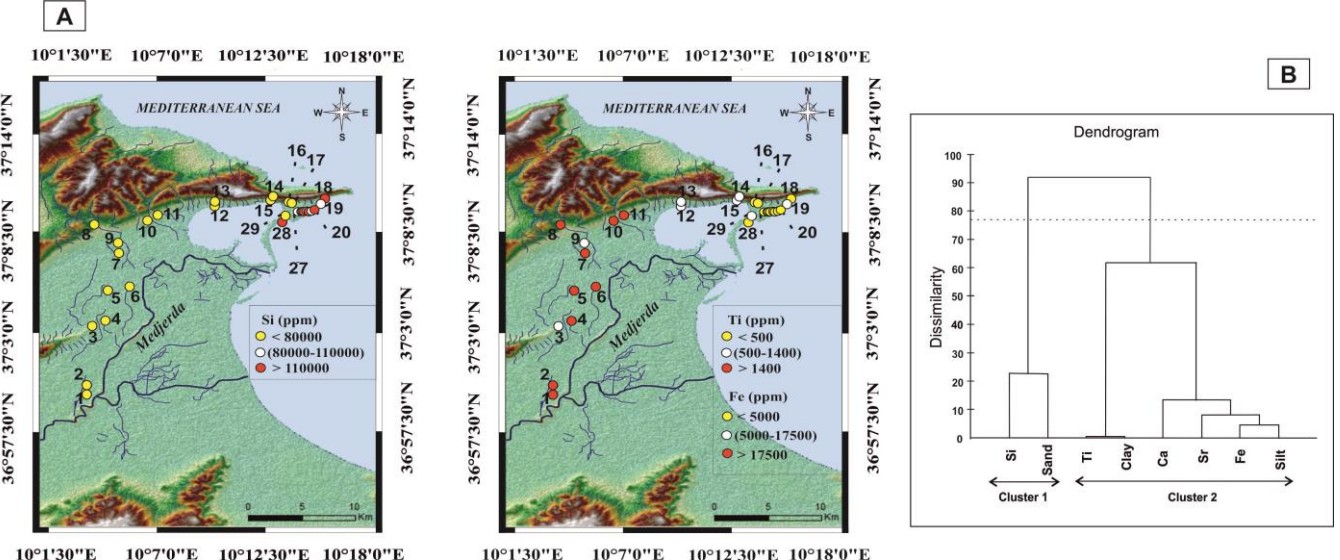

**Figure 7: 7A: Topographic maps of the Iron (Fe), Titanium (Ti) and Silicon (Si) contents in the coastal plain of Medjerda and around Ghar el Melh lagoon (Created by Radewane Hout using Arcgis). 7B: Cluster analysis of surface sediments around Ghar el Melh lagoon.**

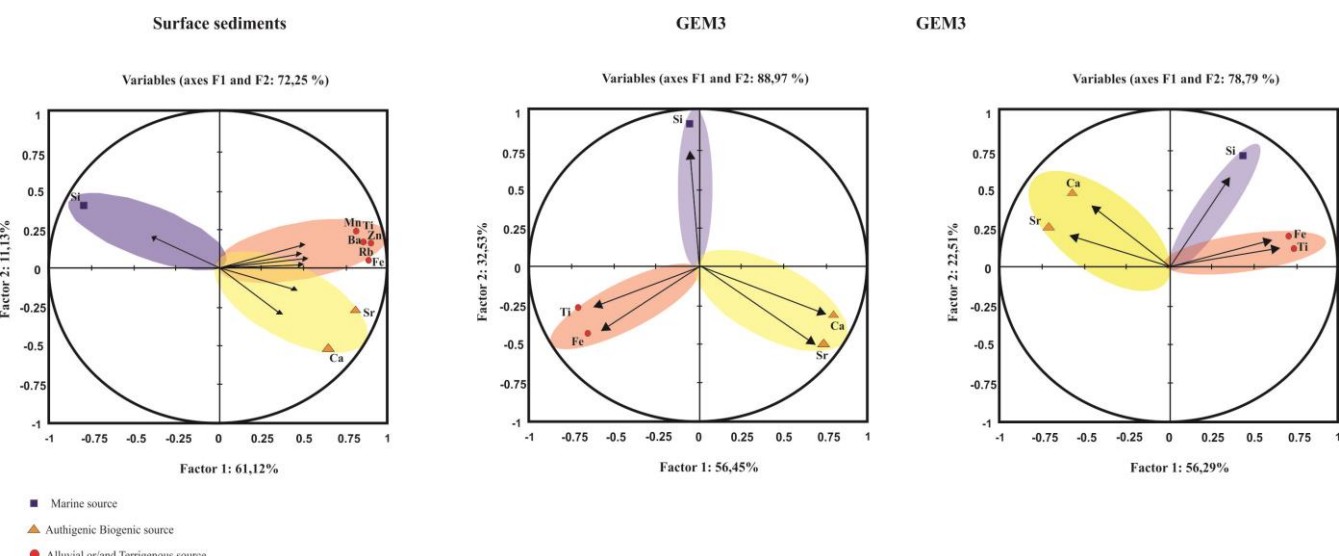

**Figure 8: The different surface sediment sources from the Medjerda watershed and Ghar el Melh lagoon obtained by using a statistic model "A Principal component analysis" (PCA).**

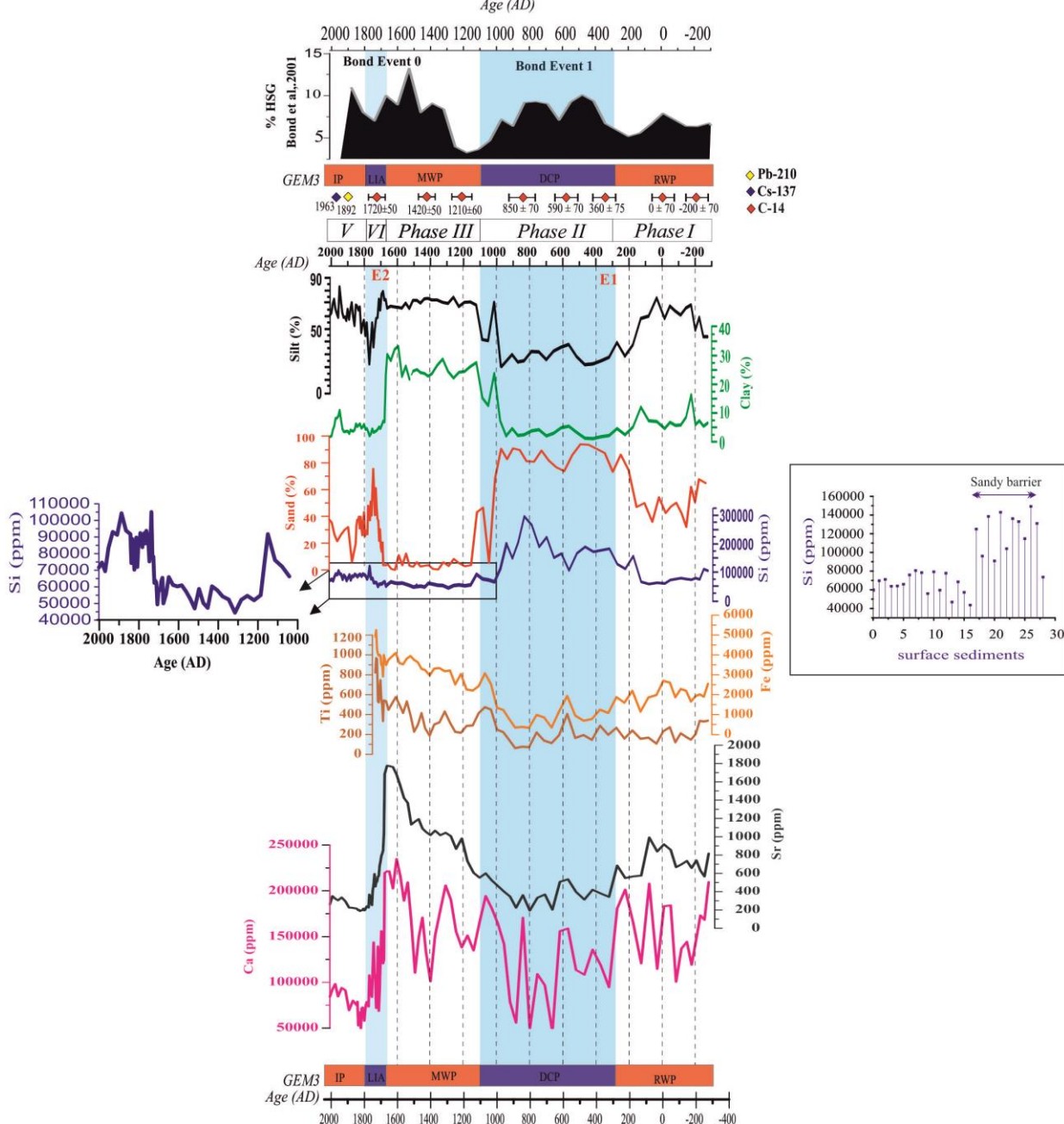

**Figure 9: Evolution of grain size population compared to the results obtained from the XRF records (Iron, Calcium, Silicon, Titanium and Strontium) distributing in the GEM3 core, vs. age from -275 till 2012 AD. The two blue bands indicate a period of the Lagoon opening.**

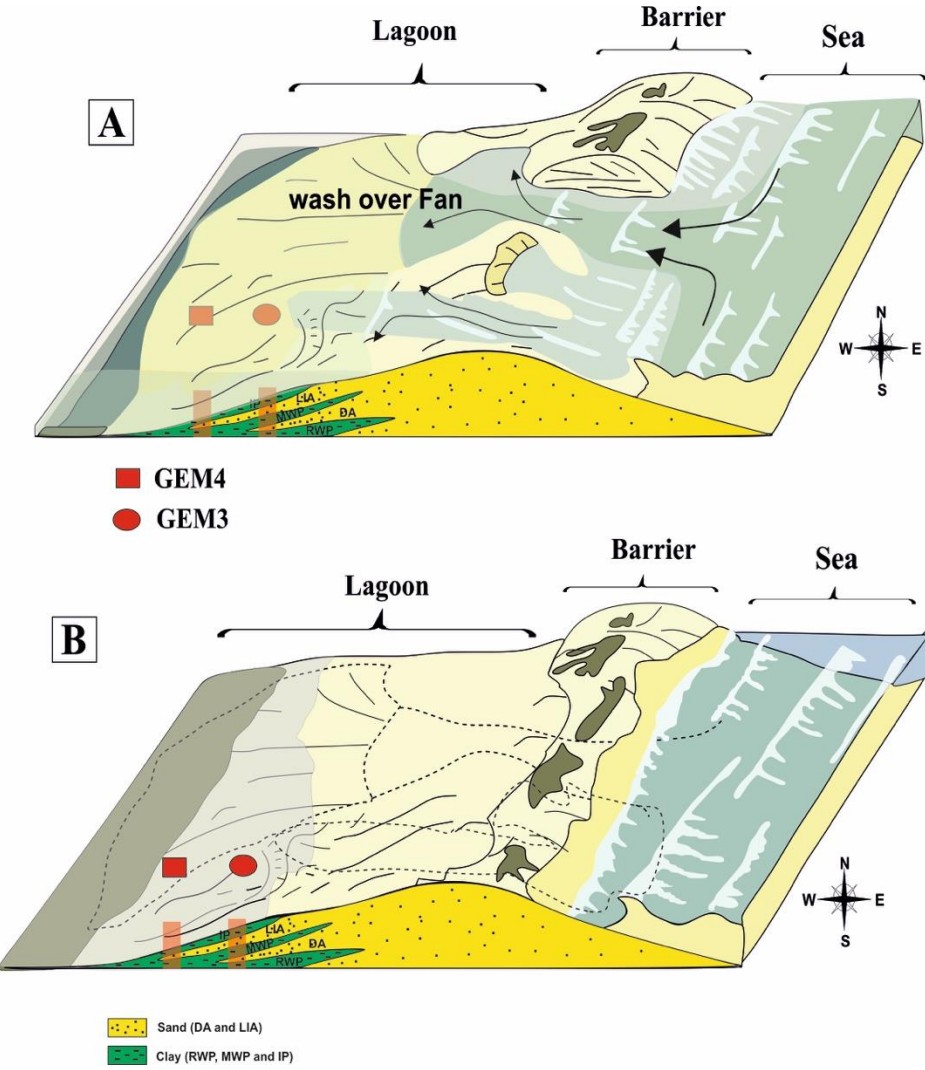


**Figure 10: 3D diagram of the lagoon Ghar el Melh evolution. (A) Opening of the sandy lido and deposition of the washover in the lagoonal sediment. (B) Reclosure of the lagoon.**



**Tables**

**Tab.1.** **¹⁴C data for shells from GEM3. The model ages were determined by the Oxcal. The ¹⁴C ages are calculated and calibrated using the curve of Marine 13 calibration with reservoir age.**

| Labo code | Mollusk used | Depth (cm) | δ¹³C (‰) | ¹⁴C ages (BP) | ¹⁴C ages (Cal AD) (One Sigma ranges) [Start: end] |
|---|---|---|---|---|---|
| SacA 42679 | *Cerastoderma glaucum* | 42 | -0.8 | 565±30 | [cal AD 1692: cal AD 1765] |
| SacA 42685 | *Cerastoderma glaucum* | 54 | 1.5 | 930±30 | [cal AD 1399: cal AD 1451] |
| SacA 42683 | *Cerastoderma glaucum* | 60 | 0.2 | 1195±3 | [cal AD 1186: cal1260] |
| SacA 42681 | *Cerastoderma glaucum* | 70 | 0.2 | 1535±30 | [cal AD 811: cal AD 902] |
| SacA 42680 | *Cerastoderma glaucum* | 76 | 1.4 | 1830±30 | [cal AD 550: cal AD 630] |
| SacA 42686 | *Cerastoderma glaucum* | 80 | 0.6 | 2015±30 | [cal AD 332: cal AD 424] |
| SacA 42684 | *Cerastoderma glaucum* | 88 | -1.1 | 2350±30 | [cal BC 67: cal AD 33] |
| SacA 42682 | *Cerastoderma glaucum* | 93 | -1.4 | 2490±30 | [cal BC 267: cal BC 144] |

**Tab.2.** **¹⁴C data for shells from GEM4. The model ages were determined by the Oxcal. The ¹⁴C ages are calculated and calibrated using the curve of Marine 13 calibration with reservoir age.**

| Labo code | Mollusk used | Depth (cm) | δ¹³C (‰) | ¹⁴C ages (BP) | ¹⁴C ages (Cal AD) (One Sigma ranges) [Start: end] |
|---|---|---|---|---|---|
| SacA42676 | *Cerastoderma glaucum* | 40 | -3.7 | 490±30 | [cal AD 1807: cal AD 1906] |
| SacA42675 | *Cerastoderma glaucum* | 45 | 1.8 | 615±30 | [cal AD 1652: cal AD 1712] |
| SacA42673 | *Cerastoderma glaucum* | 55 | 0.3 | 1181±30 | [cal AD 786: cal AD 1344] |
| SacA42672 | *Cerastoderma glaucum* | 72 | -2.9 | 2545±30 | [cal BC 332: cal BC 225] |
| SacA42674 | *Cerastoderma glaucum* | 85 | -4.3 | 2660±30 | [cal BC 431: cal BC 353] |
| SacA42678 | *Cerastoderma glaucum* | 101 | -6.6 | 2625±30 | [cal BC 399: cal BC 339][a] |
| SacA42677 | *Cerastoderma glaucum* | 110 | -4.8 | 2725±30 | [cal BC 512: cal BC 400] |

[a]Age inversion

**Tab.3 .$^{14}$C age of recent pre-bomb mollusk samples in GEM4 core. The reservoir age R(t) and ΔR were calculated using the $^{210}$Pbex date**

| Labo code | $^{210}$Pb age (AD) | $^{14}$C year (BP) | Tree-ring $^{14}$C age (BP) IntCall 13 | Reservoir age R(t) (year) | Model age (Marine 4 curve) | Δ R (year) |
|-----------|---------------------|--------------------|----------------------------------------|----------------------------|-----------------------------|------------|
| SacA44506 | 1845 | 450±30 | 114±8 | 336 | 488±23 | -38 |