# Peer review of "Extreme marine events revealed by lagoonal sedimentary records in Ghar el Melh during the last 2500 years in the northeast of Tunisia."

_Natural Hazards and Earth System Sciences, 2020_

## Referee Comment (RC1) · Anonymous Referee #1 · 24 Nov 2020

The manuscript presents the sedimentological and geochemical investigation of lagoon deposits in Tunisa, aimed at the recognition of extreme sea surges of the last three millennia. Considering the NHESS focus, the topic is of potential interest for the journal community. However, I see serious flaws in the presentation and discussion of the results. I try to synthetize here what I consider to be the major weak points that do not allow the manuscript to be published in the present form.

Settings The setting lacks of basic information such as: the lithology of the bedrock surrounding the lagoon and forming the Medjerda river catchment; the Medjerda water and solid discharges (average and during floods); the amplitude of local tides and the

existence of long-shore littoral currents; present wave heights and sea surge during severe storms.

Methods There is no indication of the method adopted for coring GEM3 and GEM4 ("piston core" is very general, e.g.: were they manually-operated? Which was the diameter of the core?) and for the collection of surface samples. Concerning these latter, no information is provided on how shallow they were (few centimetres? Few decimeters?), their geomorphological position in the landscape (river bed? Floodplain? Beach? Dunes? Etc.) and if they were collected from present-day soil horizons (meaning they may be slightly weathered sediment). No indication is provided on the grain-size classification applied.

Calibration of 14C ages Inaccuracy inherent to radiocarbon dating is not expressed in Table 1 and 2. Probability of the calibrated ages as 1 or 2 sigma is not reported. Altogether, these errors are not discussed when integrating the 210Pb and 137Cs chronologies and estimating sedimentation rates. This raises concerns on the effective existence of the hypothesized time-correlations between the datasets and the Bond events.

Discussion and conclusions A prominent conceptual inadequacy is the apparent non-consideration by the authors of the so-called Walther's Law (Walther, 1894). This basic law of stratigraphy states that any vertical progression of facies is the result of a succession of depositional environments that are laterally juxtIaposed to each other (Lopez, 2015, in Encyclopedia of Scientific Dating Methods). This implies that changes in the lithofacies characteristics and grain-size distribution in the two cored locations may be autocyclic, due to, e.g., the migration of the barrier islands, shifting of tidal inlets and channels, progradation of lagoonal deltas. Vertical changes in sediment characteristics can thus occur without the intervention of "extreme" climatic events. This depotentiates the conclusions of the authors, that appear not fully supported by the data. It further shows that alternative interpretations should be considered.

[Figure]

English language Poor English language recurrently hinders the comprehension of the concepts presented in the text.

---

## Referee Comment (RC2) · Anonymous Referee #2 · 25 Nov 2020

The manuscript "Extreme marine events revealed by lagoonal sedimentary records in Ghar el Melh during the last 2500 years in the northeast of Tunisia" by Balkis Samah Kohila and colleagues focuses on paleo-extreme events and paleo-environmental changes of the Northeastern part of Tunisia during the Late Holocene.

I do have some major recommendations to improve this paper:

The first issue is the chronology. Figure 7 clearly illustrates the uncertainty linked with the age model. The core GEM 4 is badly dated (where is the table with the details of the radiocarbon dates?). A chronological gap of ∼1400 years in the middle of the core hinders any calculation of a secure age-depth model, and may suggest an important

chronological issue. The authors must correct this point (more radiocarbon dates in the middle part of the core), otherwise, they must discuss this chronological uncertainty in the manuscript. As it stands, the core GEM 4 is too "subject to caution" to be really used here as evidence. The part "4.2.1. Age model" is clearly insufficient to answer to this issue.

The second problem is the lack of objective analyses to probe what the authors claim. Statistical analyses must be considered here. The PCA (Fig. 2) is a good start but other tests must be used. What is the software used for the PCA? With which parameters? Why the authors did not apply the same PCA on all the data from the cores? Why were only the "surface sediment sources" included in the matrix? The authors must develop this analytical part to probe their conclusions and, mostly, they must use all the data from their cores, not only the surface deposits.

The discussion is too weak to really be "a discussion". This part only summarizes the results, with auto-citations, and does not compare or integrate the data in a wider perspective (climate shifts, human impacts, etc...). This part must be rewritten and must integrate more references, more comparisons, more "other hypotheses". As it stands, we have the bad impression that the authors take their results as evidences and do not feel the need to compare or integrate their findings with what has been previously published on this subject. More caution is needed here.

The paper presents interesting data. Nonetheless, there are a number of problems that must be addressed before publication.

---

## Author Comment (AC1) · 26 Jan 2021

The authors would like to thank the Anonymous Referee #1 for his valuable comments and suggestions, they will be seriously taking into consideration and corresponding corrections will be made in the next version of the manuscript. However, we present some clarification and answers (R) to his questions (Q) in the following text : Settings : Q1 :The setting lacks of basic information such as: the lithology of the bedrock surrounding the lagoon and forming the Medjerda river catchment; the Medjerda water and solid discharges (average and during floods); the amplitude of local tides and the existence of long-shore littoral currents; present wave heights and sea surge during

severe storms. R1 : We will take into consideration your comment and we will reformulate this paragraph in the next version as following : This lagoon is directly limited in the north side by a mountain range called "Jbel Nadhour" (325 m). This mountain is composed of a marine Pliocene material (Figure 1) represented by sandstone sediments. The lagoon is bordered in the west and south side by recent quaternary marshy grounds formed by clay and silt sediments. While in the eastern side, it is separated from the sea by a sandy barrier, with a local opening (El Boughaz) allowing a permanent hydraulic communication (Oueslati et al., 2006). This sandy barrier was formed by a littoral drift oriented from the North-East to the South-West. Over time, the delta of Medjerda was distinguished by a high flood peak and a high interannual discharge variability. The Medjerda water storage shows 22% of the country's renewable water resources. This river's average sediment yield is about 10 g/l and it is characterized by an annual average flow of 30 m3/s and reached 3500 m3/s in the exceptional flood of March 1973 where solid discharge reached 100 g/l. In the Gulf of Tunis, the mean amplitude of semi-diurnal micro-tidal activities measures between 12 and 30 cm (El Arrim, 1996; Saïdi et al., 2012). According to Oueslati, 1993, the amplitude of the tidal range in this region was estimated of around 35 cm. The coastal environment of Gulf of Tunis was exposed to natural erosion processes provoked by waves, tides, and periodic storm surges. This erosion is also due to the impact of the longshore coastal drift from the SE to NW direction. Methods : Q2 : There is no indication of the method adopted for coring GEM3 and GEM4 ("piston core" is very general, e.g.: were they manually-operated? Which was the diameter of the core?) and for the collection of surface samples. Concerning these latter, no information is provided on how shallow they were (few centimetres? Few decimeters?), their geomorphological position in the landscape (river bed? Floodplain? Beach? Dunes? Etc.) and if they were collected from present-day soil horizons (meaning they may be slightly weathered sediment). No indication is provided on the grain-size classification applied. R2 : We will take into consideration your comment and we will reformulate this paragraph in the next version as following : Two piston cores were manually collected in 2012 in the Northeast of Ghar el Melh

lagoon. These cores are 126 cm (GEM4) and 98 cm (GEM3) in length and 10 cm in diameter (Figure 1). They were manually sampled according a transect East West into the lagoon (âĹij200m from the sandy barrier for GEM3 and âĹij400m for GEM4). 29 surface sediment samples of around 20 to 30g were collected from present-day soil horizons from the Medejerda watershed to the littoral area (beaches and dunes). The particle sizes obtained are classified according to Folk and Ward (1957) into three categories (clay $\Phi < 2\mu$m, silt $2\mu$m $<\Phi < 63\mu$m and sand $\Phi > 63\mu$m). Q3 : Calibration of 14C ages Inaccuracy inherent to radiocarbon dating is not expressed in Table 1 and 2. Probability of the calibrated ages as 1 or 2 sigma is not reported. Altogether, these errors are not discussed when integrating the 210Pb and 137Cs chronologies and estimating sedimentation rates. This raises concerns on the effective existence of the hypothesized time-correlations between the datasets and the Bond events.

R3 : We will take into consideration your comment. Table 1 N° Labo code Mollusk used Depth (cm) $\delta$13C (‰ 14C ages (BP) 14C ages (Cal AD) (One Sigma ranges) [Start: end] 1 Sac A 42679 Cerastoderma glaucum 42 -0.8 565$\pm$30 [cal AD 1692: cal AD 1765] 2 Sac A 42685 Cerastoderma glaucum 54 1.5 930$\pm$30 [cal AD 1399: cal AD 1451] 3 Sac A 42683 Cerastoderma glaucum 60 0.2 1195$\pm$30 [cal AD 1186: cal AD 1260] 4 Sac A 42681 Cerastoderma glaucum 70 0.2 1535$\pm$30 [cal AD 811: cal AD 902] 5 Sac A 42680 Cerastoderma glaucum 76 1.4 1830$\pm$30 [cal AD 550: cal AD 630] 6 Sac A 42686 Cerastoderma glaucum 80 0.6 2015$\pm$30 [cal AD 332: cal AD 424] 7 Sac A 42684 Cerastoderma glaucum 88 -1.1 2350$\pm$30 [cal BC 67: cal AD 33] 8 Sac A 42682 Cerastoderma glaucum 93 -1.4 2490$\pm$30 [cal BC 267: cal BC 144]

Table 2 N° Labo code Mollusk used Depth (cm) $\delta$13C (‰ 14C ages (BP) 14C ages (Cal AD) (One Sigma ranges) [Start: end] 1 Sac A 42676 Cerastoderma glaucum 40 -3.7 490$\pm$30 [cal AD 1807: cal AD 1906] 2 Sac A 42675 Cerastoderma glaucum 45 1.8 615$\pm$30 [cal AD 1652: cal AD 1712] 3 Sac A 42673 Cerastoderma glaucum 55 0.3 1181$\pm$30 [cal AD 786: cal AD 1344] 4 Sac A 42672 Cerastoderma glaucum 72 -2.9 2545$\pm$30 [cal BC 332: cal BC 225] 5 Sac A 42674 Cerastoderma glaucum 85 -4.3

2660±30 [cal BC 431: cal BC 353] 6 Sac A 42678 Cerastoderma glaucum 101 -6.6 2625±30 [cal BC 399: cal BC 339]a 7 Sac A 42677 Cerastoderma glaucum 110 -4.8 2725±30 [cal BC 512: cal BC 400] aAge inversion Table 3 Labo code 210Pb age (AD) 14C year (BP) Tree-ring 14C age (BP) IntCall 13 Reservoir age R(t) (year) Model age Marine 04  R (year) Sac A 44506 1845 450±30 114±8 336 488±23 -38±20

R : In order to determine a mean value to be used with marine calibration curves, more studies are needed and our ∆R should be taken with prudence. The eight and seven shells collected respectively from GEM3 and GEM4 cores were calibrated using the oxcal 4 program with a ∆R value of − 38 ± 20 years. The results are described in Tables 1 and 2 with an error range of $1\sigma$. For the GEM4 core, we note that sample n°6 must be a remobilized material from older sediments since sample n°7 taken from a lower sedimentary level has a younger age. R : In order to determine the ∆R, these errors of 14C were discussed when integrating the 210Pb and 137Cs chronologies and the comparison was done.

Discussion and conclusions : Q : A prominent conceptual inadequacy is the apparent nonconsideration by the authors of the so-called Walther's Law (Walther, 1894). This basic law of stratigraphy states that any vertical progression of facies is the result of a succession of depositional environments that are laterally juxtlaposed to each other (Lopez, 2015, in Encyclopedia of Scientific Dating Methods). This implies that changes in the lithofacies characteristics and grain-size distribution in the two cored locations may be autocyclic, due to, e.g., the migration of the barrier islands, shifting of tidal inlets and channels, progradation of lagoonal deltas. Vertical changes in sediment characteristics can thus occur without the intervention of "extreme" climatic events. This depotentiates the conclusions of the authors, that appear not fully supported by the data. It further shows that alternative interpretations should be considered. R : In the next version, we will try to reformulate this discussion and append a paragraph Âń 5.1. Site sensitivity to overwash deposits Âż. We will try also to integrate more references and hypotheses. Generally and from a geological point of view, I totally agree

with you about the importance of Walther's law especially in stratigraphy. But in our case, we have a vertical facies formed by fine sediments interbedded by just two sandy layers. The presence of sand layers and a discontinuity contact is interpreted as a result of intense overwash events. Through the study of the GEM3 and the GEM4 cores, our objective was to identify marine submersion deposits in a general way, through their geochemical and sedimentological characteristics, without making a distinction between storm and tsunami deposits. The entrance of marine sandy materials into the lagoon cannot be explained by a migration of the barrier islands. Indeed, we could imagine that on this time scale (2000 years), the displacement of the sandy barrier is progressive with the increase of sea level. In this case, our sedimentary archives should have presented a gradual increase in the percentage of marine sand towards more recent periods. This is not observed. The observation of two discrete sandy deposits can only be explained by two exceptional events. . The same deduction can also be used regarding the progradation of the Medjerda delta. This process is also progressive and we should have seen a progressive trend in the evolution of our facies. This is also not observed. We interpret our marine sand deposits with a sharp contact at their base by exceptional events and not by progressive processes.

Q : Poor English language recurrently hinders the comprehension of the concepts presented in the text. R : The English will be revised by a native.

---

## Author Response (AR1)

**Response to the reviews including a list of all relevant changes made in the manuscript**

**For the Referee #1**

The authors would like to thank the Anonymous Referee #1 for his valuable comments and suggestions, they will be seriously taking into consideration and corresponding corrections will be made in the next version of the manuscript. However, we present some clarification and answers (R) to his questions (Q) in the following text :

Settings :

Q1 :The setting lacks of basic information such as: the lithology of the bedrock surrounding the lagoon and forming the Medjerda river catchment; the Medjerda water and solid discharges (average and during floods); the amplitude of local tides and the existence of long-shore littoral currents; present wave heights and sea surge during severe storms.

R1: We have formulated this paragraph in the new version as following :

[revised manuscript text omitted]

Q3 : Calibration of $^{14}$C ages Inaccuracy inherent to radiocarbon dating is not expressed in Table 1 and 2. Probability of the calibrated ages as 1 or 2 sigma is not reported. Altogether, these errors are not discussed when integrating the $^{210}$Pb and $^{137}$Cs chronologies and estimating sedimentation rates. This raises concerns on the effective existence of the hypothesized time-correlations between the datasets and the Bond events.

R3 : We have taken your comments into consideration as following :

Table 1

| N° | Labo code | Mollusk used | Depth (cm) | δ$^{13}$C (‰) | $^{14}$C ages (BP) | $^{14}$C ages (Cal AD) (One Sigma ranges) [Start: end] |
|---|---|---|---|---|---|---|
| 1 | Sac A 42679 | *Cerastoderma glaucum* | 42 | -0.8 | 565±30 | [cal AD 1692: cal AD 1765] |
| 2 | Sac A 42685 | *Cerastoderma glaucum* | 54 | 1.5 | 930±30 | [cal AD 1399: cal AD 1451] |
| 3 | Sac A 42683 | *Cerastoderma glaucum* | 60 | 0.2 | 1195±30 | [cal AD 1186: cal AD 1260] |
| 4 | Sac A 42681 | *Cerastoderma glaucum* | 70 | 0.2 | 1535±30 | [cal AD 811: cal AD 902] |
| 5 | Sac A 42680 | *Cerastoderma glaucum* | 76 | 1.4 | 1830±30 | [cal AD 550: cal AD 630] |
| 6 | Sac A 42686 | *Cerastoderma glaucum* | 80 | 0.6 | 2015±30 | [cal AD 332: cal AD 424] |
| 7 | Sac A 42684 | *Cerastoderma glaucum* | 88 | -1.1 | 2350±30 | [cal BC 67: cal AD 33] |
| 8 | Sac A 42682 | *Cerastoderma glaucum* | 93 | -1.4 | 2490±30 | [cal BC 267: cal BC 144] |

Table 2

| N° | Labo code | Mollusk used | Depth (cm) | δ$^{13}$C (‰) | $^{14}$C ages (BP) | $^{14}$C ages (Cal AD) (One Sigma ranges) [Start: end] |
|---|---|---|---|---|---|---|
| 1 | Sac A 42676 | *Cerastoderma glaucum* | 40 | -3.7 | 490±30 | [cal AD 1807: cal AD 1906] |
| 2 | Sac A 42675 | *Cerastoderma glaucum* | 45 | 1.8 | 615±30 | [cal AD 1652: cal AD 1712] |
| 3 | Sac A 42673 | *Cerastoderma glaucum* | 55 | 0.3 | 1181±30 | [cal AD 786: cal AD 1344] |
| 4 | Sac A 42672 | *Cerastoderma glaucum* | 72 | -2.9 | 2545±30 | [cal BC 332: cal BC 225] |
| 5 | Sac A 42674 | *Cerastoderma glaucum* | 85 | -4.3 | 2660±30 | [cal BC 431: cal BC 353] |
| 6 | Sac A 42678 | *Cerastoderma glaucum* | 101 | -6.6 | 2625±30 | [cal BC 399: cal BC 339][a] |
| 7 | Sac A 42677 | *Cerastoderma glaucum* | 110 | -4.8 | 2725±30 | [cal BC 512: cal BC 400] |

[a] Age inversion

Table 3

| Labo code | $^{210}$Pb age (AD) | $^{14}$C year (BP) | Tree-ring $^{14}$C age (BP) IntCall 13 | Reservoir age R(t) (year) | Model age Marine 04 | Δ R (year) |
|---|---|---|---|---|---|---|
| Sac A 44506 | 1845 | 450±30 | 114±8 | 336 | 488±23 | -38±20 |

R : In order to determine a mean value to be used with marine calibration curves, more studies are needed and our ΔR should be taken with prudence. The eight and seven shells collected respectively from GEM3 and GEM4 cores were calibrated using the oxcal 4 program with a ΔR value of − 38 ± 20 years. The results are described in Tables 1 and 2 with an error range of 1σ. For the GEM4 core, we note that sample n°6 must be a remobilized material from older sediments since sample n°7 taken from a lower sedimentary level has a younger age.

R : In order to determine the ΔR, these errors of $^{14}$C were discussed when integrating the $^{210}$Pb and $^{137}$Cs chronologies and the comparison was done. From ligne 180 to the ligne 183.

Discussion and conclusions :

Q : A prominent conceptual inadequacy is the apparent nonconsideration by the authors of the so-called Walther's Law (Walther, 1894). This basic law of stratigraphy states that any vertical progression of facies is the result of a succession of depositional environments that are laterally juxtIaposed to each other (Lopez, 2015, in Encyclopedia of Scientific Dating Methods). This implies that changes in the lithofacies characteristics and grain-size distribution in the two cored locations may be autocyclic, due to, e.g., the migration of the barrier islands, shifting of tidal inlets and channels, progradation of lagoonal deltas. Vertical changes in sediment characteristics can thus occur without the intervention of "extreme" climatic events. This depotentiates the conclusions of the authors, that appear not fully supported by the data. It further shows that alternative interpretations should be considered.

R : Generally and from a geological point of view, I totally agree with you about the importance of Walther's law especially in stratigraphy. But in our case, we have a vertical facies formed by fine sediments interbedded by just two sandy layers. The presence of sand layers and a discontinuity contact is interpreted as a result of intense overwash events. Through the study of the GEM3 and the GEM4 cores, our objective was to identify marine submersion deposits in a general way, through their geochemical and sedimentological characteristics, without making a distinction between storm and tsunami deposits. The entrance of marine sandy materiels into the lagoon cannot be explained by a migration of the barrier islands. Indeed, we could imagine that on this time scale (2000 years), the displacement of the sandy barrier is progressive with the increase of sea level. In this case, our sedimentary archives should have presented a gradual increase in the percentage of marine sand towards more recent periods. This is not observed. The observation of two discrete sandy deposits can only be explained by two exceptional events. . The same deduction can also be used regarding the progradation of the Medjerda delta. This process is also progressive and we should have seen a progressive trend in the evolution of our facies. This is also not observed. We interpret our marine sand deposits with a sharp contact at their base by exceptional events and not by progressive processes.

The discussion was been formulated in the new version as following :

5.1 Site sensitivity to overwash deposits

The sensitivity of the site to the overwash deposits can result from several factors such as barrier elevation, sediment supply, inlet, and a change in sea level (Donnelly et al., 2004; Scileppi and Donnelly, 2007; Dezileau et al., 2016). Generally, an increase in sea level produces a moving of the barrier landward. Thus, the highest number of sand layers in a sediment core can be the consequence of a simple sea-level change. In the Mediterranean Sea and especially during the last 5000 years, the sea level has stayed more or less stable (< 2 m, Pirazzoli, 1991; Lambeck and Bard, 2000). Studies concerning Holocene sea-level fluctuations along the Tunisian coast remain have suggested a stabilization in relative sea level during the last 6000 years (Jedoui et al., 1998), which is very small and probably not enough to completely change the deposition environment of the Ghar el Melh lagoon. Furthermore, sedimentation in the two cores commenced 2500 years ago, on this short period, the influence of the seal-level change has not drastically affected sedimentation in the center of lagoon. Cores GEM3 and GEM4 present generally a sedimentary sequence dominated by fine-grained sediments (clay and silt), suggesting that the lagoon of Ghar el Melh has succeeded in keeping a low-energy environment during the last 2500 years. The proxies applied in this work (granulometry and geochemistry), as well as the sedimentary indicators (discontinuity contact), showed that all sandy coarse peaks present within the GEM3 and the GEM4 cores were deposited through marine high-energy events and not by a gradual change in sea-level. The presence of a nearby inlet may increase the sensitivity of a particular area to storm-induced deposition. It allows for storm energy to more easily penetrate into the back-barrier area, letting a lesser storm with lower wave surge transport coarse sediment into the back barrier. If a large inlet had existed over a long period and had provided a ready conduit for sand from the Gulf of Tunis to the Ghar el Melh lagoon, this would have been reflected in the cores (deposition of thick sand layer over a long period of time). However, no evidence of such active tidal connection lasting a long time is found in sedimentological and geochemical data for the past 2500 years. Granulometric and Geochemical observations on the GEM3 and the GEM4 cores show a sedimentary sequence dominated by silty-clay deposits and interrupted over time by some allochthonous coarse materials. To determinate the source of these sandy layers, we compared their geochemical attributes with those from surface samples. This allowed us to establish whether these deposits had a continental (river floods) or marine origin (storms and tsunamis). The two sand deposits present in the GEM3 and GEM4 cores have a geochemical correlation with marine coastal surface samples; both show enrichment with Si and depletion with Fe and Ti, which reveals a marine source for these high-energy deposits. The results demonstrate that Ghar el Melh lagoon has been confronted with different episodes of marine submersion during the last 2500 years.

5.2 Extreme events and paleoevironement changes

In this part, we have added a new paragraph (from ligne 274 to the ligne 283) as following:

This sand deposit may have had another origin. Indeed, this deposit was dated at around cal AD 332+/-30. This period coincides with the tsunami event of 365 AD. This extreme event was generated by an earthquake of 8.3 magnitudes and is supposed to have been the powerful ever in the regions of the Eastern Mediterranean. From a numerical modeling, Pararas-Carayannis and Mader, (2010) indicate that the 365 AD tsunami has heavily affected coastal areas throughout the Eastern Mediterranean region; Palestine, South Asia Minor, Cyprus, the Nile Delta, Careen, Apollonia. In the central part of the Mediterranean region the cities of Eastern Sicily, the coastline of Calabria, and the islands of Aiolou have been affected (Pararas-Carayannis, 2011). In Tunisia, the recent archeological discovery of the immersed city of Neapolis in northern Golf of Hammamet in 2017 suggest the occurrence of a tsunami in 365 AD (Aounallah and Fantar, 2006; National Heritage Institute of Tunisia, 2017). We can thus hypothesize that our sand deposited around 365 cal AD could also be associated to this tsunami event.

Q : Poor English language recurrently hinders the comprehension of the concepts presented in the text.

R : The English has been revised by a native.

**For the Referee #2**

The authors would like to thank the Anonymous Referee #2 for his valuable comments and suggestions, they will be seriously taking into consideration and corresponding corrections will be made in the next version of the manuscript. However, we present some clarification and answers (R) to his questions (Q) in the following text :

Q1 : The first issue is the chronology. Figure 7 clearly illustrates the uncertainty linked with the age model. The core GEM 4 is badly dated (where is the table with the details of the radiocarbon dates?). A chronological gap of ~ 1400 years in the middle of the core hinders any calculation of a secure age-depth model, and may suggest an important chronological issue. The authors must correct this point (more radiocarbon dates in the middle part of the core), otherwise, they must discuss this chronological uncertainty in the manuscript. As it stands, the core GEM 4 is too "subject to caution" to be really used here as evidence. The part "4.2.1. Age model" is clearly insufficient to answer to this issue.

R1: Figure 7B determines a period of inactive deposition process or erosion so-called "a Condensed area" and not a chronological gap in the age model. Supplementary radiocarbon dates between 55 cm and 72 cm (an interval of 17 cm only) will not give more information because we have a condensed area. The GEM4 is well dated with seven radiocarbon dates. We have a 14C date every 10 cm approximatively. It is very rare to observe in the literature sedimentary archives dated every 10 cm. Thus, we will not realize more radiocarbon dates on this core in order to have a better chronological framework. This has never been done in coastal environments.

Q2 : The second problem is the lack of objective analyses to probe what the authors claim. Statistical analyses must be considered here. The PCA (Fig. 2) is a good start but other tests must be used. What is the software used for the PCA? With which parameters? Why the authors did not apply the same PCA on all the data from the cores? Why were only the "surface sediment sources" included in the matrix? The authors must develop this analytical part to probe their conclusions and, mostly, they must use all the data from their cores, not only the surface deposits.

R2 : In coastal environments, the principal component analysis (PCA) was usually performed on the sediment sampled around lagoons in order to characterize the different sources of sediments deposited in the lagoon and to determine the several poles related to potential sediment sources supplies. We choose the Mn, Ti, Zn, Ba, Rb, Fe, Sr, Ca, and Si elements due to their good detection by the mobile XRF. We established the calculation factors F1 61.12% and F2 11.13% of the geochemical dataset using the XLSTAT-2016 statistical software. The apply of the same PCA on al the data from the two cores is a good idea, but in our study, the tracing of sources should be done on recent surface sediments and not on cores deposits (Degeai et al., 2015, Gaceur et., al 2017, Affouri et al.,2017 and Khalfaoui et al,. 2019 ).

Q3 : The discussion is too weak to really be "a discussion". This part only summarizes the results, with auto-citations, and does not compare or integrate the data in a wider perspective (climate shifts, human impacts, etc: : :). This part must be rewritten and must integrate more references, more comparisons, more "other hypotheses". As it stands, we have the bad impression that the authors take their results as evidences and do not feel the need to

compare or integrate their findings with what has been previously published on this subject. More caution is needed here.

R3 : The discussion was been formulated in the new version as following :

5.1 Site sensitivity to overwash deposits

The sensitivity of the site to the overwash deposits can result from several factors such as barrier elevation, sediment supply, inlet, and a change in sea level (Donnelly et al., 2004; Scileppi and Donnelly, 2007; Dezileau et al., 2016). Generally, an increase in sea level produces a moving of the barrier landward. Thus, the highest number of sand layers in a sediment core can be the consequence of a simple sea-level change. In the Mediterranean Sea and especially during the last 5000 years, the sea level has stayed more or less stable (< 2 m, Pirazzoli, 1991; Lambeck and Bard, 2000). Studies concerning Holocene sea-level fluctuations along the Tunisian coast remain have suggested a stabilization in relative sea level during the last 6000 years (Jedoui et al., 1998), which is very small and probably not enough to completely change the deposition environment of the Ghar el Melh lagoon. Furthermore, sedimentation in the two cores commenced 2500 years ago, on this short period, the influence of the seal-level change has not drastically affected sedimentation in the center of lagoon. Cores GEM3 and GEM4 present generally a sedimentary sequence dominated by fine-grained sediments (clay and silt), suggesting that the lagoon of Ghar el Melh has succeeded in keeping a low-energy environment during the last 2500 years. The proxies applied in this work (granulometry and geochemistry), as well as the sedimentary indicators (discontinuity contact), showed that all sandy coarse peaks present within the GEM3 and the GEM4 cores were deposited through marine high-energy events and not by a gradual change in sea-level. The presence of a nearby inlet may increase the sensitivity of a particular area to storm-induced deposition. It allows for storm energy to more easily penetrate into the back-barrier area, letting a lesser storm with lower wave surge transport coarse sediment into the back barrier. If a large inlet had existed over a long period and had provided a ready conduit for sand from the Gulf of Tunis to the Ghar el Melh lagoon, this would have been reflected in the cores (deposition of thick sand layer over a long period of time). However, no evidence of such active tidal connection lasting a long time is found in sedimentological and geochemical data for the past 2500 years. -Granulometric and Geochemical observations on the GEM3 and the GEM4 cores show a sedimentary sequence dominated by silty-clay deposits and interrupted over time by some allochthonous coarse materials. To determinate the source of these sandy layers, we compared their geochemical attributes with those from surface samples. This allowed us to establish whether these deposits had a continental (river floods) or marine origin (storms and tsunamis). The two sand deposits present in the GEM3 and GEM4 cores have a geochemical correlation with marine coastal surface samples; both show enrichment with Si and depletion with Fe and Ti, which reveals a marine source for these high-energy deposits. The results demonstrate that Ghar el Melh lagoon has been confronted with different episodes of marine submersion during the last 2500 years.

5.2 Extreme events and paleoevironement changes

In this part, we have added a new paragraph (from ligne 274 to the ligne 283) as following

This sand deposit may have had another origin. Indeed, this deposit was dated at around cal AD 332+/-30. This period coincides with the tsunami event of 365 AD. This extreme event was generated by an earthquake of 8.3 magnitudes and is supposed to have been the powerful ever in the regions of the Eastern Mediterranean. From a numerical modeling, Pararas-Carayannis and Mader, (2010) indicate that the 365 AD tsunami has heavily affected coastal areas throughout the Eastern Mediterranean region; Palestine, South Asia Minor, Cyprus, the Nile Delta, Careen, Apollonia. In the central part of the Mediterranean region the cities of Eastern Sicily, the coastline of Calabria, and the islands of Aiolou have been affected (Pararas-Carayannis, 2011). In Tunisia, the recent archeological discovery of the immersed city of Neapolis in northern Golf of Hammamet in 2017 suggest the occurrence of a tsunami in 365 AD (Aounallah and Fantar, 2006; National Heritage Institute of Tunisia, 2017). We can thus hypothesize that our sand deposited around 365 cal AD could also be associated to this tsunami event.

---

## Referee Report (RR1)

[referee-annotated manuscript omitted]

---

## Author Response (AR2)

**Response to the reviews**

**For the Referee #1**

The authors would like to thank the Anonymous Referee #1 for his valuable comments and suggestions, they will be seriously taking into consideration and corresponding corrections will be made in the next version of the manuscript. However, we present some clarification and answers (R) to his questions (Q) in the following text :

Q : I do not understand why the authors have not performed statistical analyses other than one PCA. This is clearly lacking and the authors must more thoroughly investigate their claims (see my previous review). To mention the use of XL-STAT is not enough to justify robust conclusions. I agree that the authors should use a PCA of modern samples to test the sources of sediment deposits, but the authors should use further statistical tools to test the value of the data obtained. The authors must use PCA or other statistical analyses on their core deposits to test the reliability of their signal. Secondly, they must compare their present-day dataset (their PCA) with the two cores. The authors must objectively demonstrate their conclusions.

R: Thank you very much again for your comment. We applied the same PCA on all the data from the two cores. Based on these statistical analyses of all geochemical data (surface sediments and cores), three distinct sources of sediments were identified. So we can conclude that the PCA result of cores confirms the results obtained by the PCA of surface sediments. Many techniques have been developed for this purpose, but principal component analysis (PCA) is one of the oldest and most widely used statistical techniques in environmental geochemistry. This multivariate approach is used to reduce a large number of variables that result from XRF analysis. PCA was applied to elements to distinguish the different sediment sources of surface sediments and link them to the geochemical processes or proprieties.

**For the Referee #Raphaël Paris**

The authors would like to thank Professor Raphaël Paris for his valuable comments and suggestions, they will be seriously taking into consideration and corresponding corrections will be made in the next version of the manuscript. However, we present some clarification and answers (R) to his questions (Q) in the following text:

Q: It is really necessary to revise English (by a native).

R: The english have been revised by a native.

Q: The bibliography is not up-to-date or sometimes not appropriate (e.g. lines 41 or 50).

R: Done

- Section 2.1 could be renamed "Geological and geomorphological setting"

R: Done

Q: In section 3.2 you should clearly distinguish analyses using a XRF handheld device (bulk samples), and XRF core scanner analyses.

R: Done. These analyses were realized by an XRF handheld device

Q: In the results, I would present first the XRF and grain size analyses on cores, and then the possible sediment sources. Please consider inverting sections 4.1 and 4.2.

R: Done.

Q: Lines 144-155 should moved to the beginning of the section, and links should be proposed between the Ti, Si, and Fe trends, and the lithology of the catchment area.

R: Done.

Q: In section 4.2. you should present first the characteristics of the cores, and then the age model (thus inverting 4.2.1 and 4.2.2).

R: Done.

Q: Does the age model takes into account the "event" deposits in terms of thickness and depth correction?

R: Yes, the age model takes into account every event deposits in terms of thickness and depth correction

Q: The statistical analysis (e.g. PCA) must be carried out FIRST on the XRF downcore data, which is of better quality and more reliable than data from mobile XRF.

R: Done. A statistical analysis PCA has been applied on the XRF downcore data and, they confirm the result obtained by the surface sediment with the three poles.

Q: The XRF data should be presented not only as single element curves (in cps rather than ppm?), but also as ratios of elements (e.g. Ca/Ti on normalized values, or ln(Ca/Ti)). PCA analysis should be run using normalized values. Then the sedimentary facies should be compared with the main components in order to identify the sediment end-members (e.g. marine vs continental).

R : The elemental concentrations obtained in this work using the hand-held Nitron XL3t (not by the XRF core scanner analyses) are expressed automatically in ppm or percentage values.

In order to interpret such datasets, methods are required to drastically reduce their dimensionality in an interpretable way, such that most of the information in the data is preserved. Many techniques have been developed for this purpose, but principal component analysis (PCA) is one of the oldest and most widely used statistical technique in environmental geochemistry. This multivariate approach is used to reduce a large number of variables that result from XRF analysis. PCA was applied to elements in order to distinguish the different sediment sources of surface sediments and link them to the geochemical processes or proprieties.

Q: The cores are poorly described. You defined three units, and later proposed 5 phases in the discussion. How are these units defined? We really need a more detailed description of the cores (main units, subunits, sedimentary facies, discontinuities, sorting, vertical grading, etc.).

R: We have considered your comments and, we have formulated this paragraph in the new version as following

The GEM3 and GEM4 sediment cores of 97 and 126 cm long show visual variation in the sediment composition. Lithological description of these two cores highlighted five distinct sedimentary facies (Figure 2):

The first unit (1), situated between 126-67 cm in GEM4 and between 97-85 cm in GEM3, is composed generally of a light grey silt layer and shells. At the base (the last 3 cm for the GEM3 and the last 13 cm for the GEM4) the lithological composition of this unit is characterized by a very thin fine sand. For the GEM3, the transition between unit (1) to unit (2) is defined by a sharp contact (Figure 2).

The second unit (2), situated between 85-63 cm in GEM3 and between 66-60 cm in GEM4, is typically composed of a light grey sand with a combination of shell fragments and siliciclastic grains. It is probably related to marine incursion and washover event during an intense event such as a storm or tsunami.

The third unit (3), situated between 60-36 cm in the GEM3 and between 63-30 cm in the GEM4, is composed mainly of grey silt and shells. In fact, the transition between this unit and the unit subjacent is defined by a discontinuity contact (Figure 2).

The fourth unit (4), is about 10 cm think in GEM3 and 6 cm in the GEM4. This coarse layer is constituted by a mixture of shell debris and siliciclastic sand. This sand layer is usually characterized by coarse sediments with light colours and also dominated by shell fragments. This coarse grain size layer intercalated in the mud sediments indicates an "energetic" event, relative to the background sedimentation. It is probably also linked to washover event and marine incursion during an intense event such as a storm or tsunami.

The fifth unit (5) presenting a thickness of 26 cm (GEM3) and 24 cm (GEM4) and marked by a massive grey to dark clay with trace of oxidized plant roots in the last three centimeters.

Q :Then perhaps you will be able to provide some clues on the distinction between (a) storms, (b) periods of increasing storminess (successive storms), or (c) tsunami.

R: Both tsunamis and storms are considered extreme sea events and can cause short coastal flooding with high overland flow velocities. The sedimentary characteristics of tsunamis or storm deposits are almost similar (Hawkes

et al., 2007; Kortekaas and Dawson, 2007; Morton et al., 2007; Mamo et al., 2009). Hence, the distinction between these two coarse deposits is still controversial and several studies have pointed out many hypotheses regarding the diagnostic characteristics of storm or tsunami deposits (Kortekaas and Dawson, 2007; Morton et al., 2007; Tappin, 2007; Engel and Brückner, 2011; Sakuna-Schwartz et al., 2015). For example, Morton et al., 2007, used some sedimentological criteria to distinguish storm from tsunami deposits. According to them, the storm-originated deposits present a moderately thick sand bed composed of several sub-horizontal planar laminations organized into multiple laminates. The stratification associated with bed-load transport and abundant shell fragments organized in laminations also favors a storm origin. In contrast, the presence of internal mud laminae or mud intraclasts is stronger evidence of tsunami deposits. However, in our case, the sand bed that corresponds to the extreme event is characterized by a single homogeneous bed (6–9 cm thick) with no evident sedimentary structures (such as laminations) that correspond neither to storm nor tsunami deposits. To precise the origin of our thin coarse layer, we explored the regional historical storm's records and tsunamis data.

---

## Author Response (AR3)

**Response to the reviews**

**For the Editor #Paolo Tarolli**

The authors would like to thank the Professeur Paolo Tarolli for his valuable comments and suggestions, they will be seriously taking into consideration and corresponding corrections will be made in the next version of the manuscript. However, we present some clarification and answers (R) to his questions (Q) in the following text :

Q : Your article has been further revised by one reviewer who suggested minor changes. However, it seems that you did not reply (nor enriched the text) to the point I raised: I stressed the fact that the article should contain more mentions on hazards (the word "hazard" is just mentioned in the references), in order to meet the NHESS journal scope, otherwise the work will remain closed to a pure geology/geomorphology journal rather than our journal. Please enrich the text, both on the introduction and discussion part, with more mentions on hazards and possible advances of this research in coastal risk management.

R: Thank you very much again for your comment. We have considered your comments and, we have formulated the abstract, the introduction, and we add sentences in the discussion as following :

The abstarct :

-and identify two sediment layers that are in connection with two major historical marine submersion events. The first layer is mentioned as E1 and seems to fit with the great tsunami of 365 Cal AD. This event was marked by an increase in the coarse sediment and it is correlated for the first time with the immersed city of Neapolis in northern Golf of Hammamet in 2017 by the same tsunamis of 365 Cal AD. The other sandy layer referred to as E2 was dated from 1690 to 1760 Cal AD, and is marked by one specific sedimentological layer attributed to a marine submersion event. This layer could be associated with the 1693 tsunami event in southern Italy or an increase in extreme storm events.

The Introduction:

Add a new sentence (ligne 33): During the last century, coastal communities have become very vulnerable to many extreme events such as tsunamis, tropical storms, hurricanes, and floods (Chaumillon et al., 2017). Risks and vulnerabilities of the coastal area have recently increased, not only because of the sea-level rise the changes in climate conditions but also because of the high number of natural catastrophes disasters, and the construction of nonplanned urban areas (Cardona A., 2001; Milanés Batista et al., 2017).

-Add a new paragraph (from ligne 56): The Tunisian coast has been exposed to numerous extreme hazards (floods, storms, and tsunamis) (Rizzi et al., 2016; Zaïbi et al., 2016; Affouri et al., 2017; Khadraoui et al., 2018; Amrouni et al., 2019). During the last century, this coastal area has experienced some coastal marine storms (Zaïbi et al., 2016). Moreover, this area is also subject to tsunami events, which can especially come from the seismic source related to the tectonic activities in Southeastern Sicily e.g the immersed city of Neapolis in northern Golf of Hammamet in 2017 suggest the occurrence of a tsunami in 365 AD (Aounallah and Fantar, 2006; National Heritage Institute of Tunisia, 2017).

- Add a new paragraph:(Morton et al., 2007; Dezileau et al., 2011; Sabatier et al., 2012). This geological approach using sedimentological and geochemical analyses has been used in the French, Morocco, and Spanish coasts (Degeai et al., 2015; Dezileau et al., 2016; Khalfaoui et al., 2019) . Inversely, only a few high-resolutions studies have been conducted on the Tunisian coast.

- Add a new sentence :In this context, the present study aims to reconstruct past marine submersion events from geological archives (cores) collected from the Ghar el Melh lagoon (NE of Tunisia) using a high-resolution sedimentological and geochemical analysis.

Discussion

Add a new paragraph as following:

-In this respect, the lagoonal deposit of Ghar el Melh can provide valuable information on these aspects of the past and subsequently provide a forecast about the future. So we can suggest that the Tunisian coast is very sensitive to extreme events, especially these coastal areas are vital for Tunisia's tourism development and economy. These hazard events can expose in the future many destructions and caused significant human and economic losses. In

this fact, many managements to risk should be taken into consideration and applied by the governorate. In this fact, many managements to risk in this coastal zone should be taken into consideration and applied by the governorate. A Regional Risk Assessment methodology must be developed for the assessment of the potential impacts of climate change in the Tunisian coastal zone of the Ghar El Melh lagoon.

**For the Referee #Anonymous Referee 2**

The authors would like to thank Anonymous Referee #2 for his valuable comments and suggestions, they will be seriously taking into consideration and corresponding corrections will be made in the next version of the manuscript. However, we present some clarification and answers (R) to his questions (Q) in the following text:

Q: The revised manuscript "Extreme marine events revealed by lagoonal sedimentary records in Ghar el Melh during the last 2500 years in the northeast of Tunisia" by Balkis Samah Kohila and colleagues is better than the previous version. I particularly appreciate the use of PCAs.

To go further, I would suggest the use of cluster analyses (cluster analysis, neighbor joining analysis) to test the statistical link between all the elements and the deposits. This must be done using both sedimentological data and XRF data. All data should be transformed into z-scores to avoid any bias due to the different scales of measurement. This would strengthen the outcomes and make the manuscript more robust.

R: Thank you very much again for your comment.

A cluster anlyses of surface sediments was done (Figure 7B) and the data (ppm and %) was transformed intoo z-scores.

-Add a new paragraph as following:

**4.1 Characterization of different detrital surface sources**

To make our interpretations more vigorous, a tree diagram was generated using the statistical program XLSTAT 2021 statistical software, which is used as an additional tool to identify and test the statistical link between all the elements and the deposits using both sedimentological data and XRF data of surface sediments in the study area (Figure 7B). In the first cluster, the association of coarse fraction (Sand) with the Si is clear, suggesting that the silicone is coming from coarse marine sand inputs. However, the second cluster determines an assembly between the terrigenous elements (Ti, Fe, Sr, and Ca) and fine fractions (Silt and Sand). This difference in the origin of the terrigenous inputs in Ghar el Melh lagoon explained by the fact that, during floods events, finer sediments are coming from the Medejerda watershed whereas, at the time of marine storms, coarse marine sand inputs are from the barrier.